**Article** https://doi.org/10.1038/s41467-023-37418-8

# Active learning-assisted neutron spectro-scopy with log-Gaussian processes

**Mario Teixeira Parente** [1] ✉, **Georg Brandl** [1], **Christian Franz** [1], **Uwe Stuhr** [2], **Marina Ganeva** [1] ✉ **& Astrid Schneidewind** [1]

Neutron scattering experiments at three-axes spectrometers (TAS) investigate magnetic and lattice excitations by measuring intensity distributions to understand the origins of materials properties. The high demand and limited availability of beam time for TAS experiments however raise the natural question whether we can improve their efficiency and make better use of the experimenter's time. In fact, there are a number of scientific problems that require searching for signals, which may be time consuming and inefficient if done manually due to measurements in uninformative regions. Here, we describe a probabilistic active learning approach that not only runs autonomously, i.e., without human interference, but can also directly provide locations for informative measurements in a mathematically sound and methodologically robust way by exploiting log-Gaussian processes. Ultimately, the resulting benefits can be demonstrated on a real TAS experiment and a benchmark including numerous different excitations.

Neutron three-axes spectrometers (TAS)[1] enable understanding the origins of materials properties through detailed studies of magnetic and lattice excitations in a sample. Developed in the middle of the last century, the technique remains one of the most significant in fundamental materials research and was therefore awarded the Nobel prize in 1994[2]. It is used for investigating the most interesting and exciting phenomena of their time: the cause of different crystal structures in iron[3], unconventional superconductors[4], quantum spin glasses[5], quantum spin liquids[6], and non-trivial magnetic structures[7].

TAS are globally operated at neutron sources of large-scale research facilities and measure scattering intensity distributions in the material's four-dimensional **Q**-$E$ space, i.e., in its momentum space (**Q**) for different energy transfers ($E$)[8], by counting scattered neutrons on a single detector. However, high demand and limited availability make beam time at TAS a valuable resource for experimenters. Since, furthermore, the intensity distributions of the aforementioned excitations have an information density that strongly varies over **Q**-$E$ space due to the direction of the underlying interactions and the symmetry of the crystal, and TAS measure sequentially at single locations in **Q**-$E$ space, it is natural to think about if and how we can improve the

efficiency of TAS experiments and make better use of the experimenter's time.

In experimental workflows at TAS, there are scenarios where the intensity distribution to be measured is not known in advance and therefore a rapid overview of the same in a particular region of **Q**-$E$ space is required. So far, experimenters then decide manually how to organize these measurements in detail. In this mode, however, there is a possibility depending on the specific scenario that measurements do not provide any further information and thus waste beam time since they are placed in the so-called background, i.e., regions with either no or parasitic signal. If there were computational approaches that autonomously, i.e., without human interference, place measurements mainly in regions of signal instead of background in order to acquire more information on the intensity distribution in less time, not only the use of beam time gets optimized, but also the experimenters can focus on other relevant tasks in the meantime.

The potential of autonomous approaches for data acquisition was recognized throughout the scattering community in recent years. gpCAM, for example, is an approach that is also based on GPR and applicable to any scenario in which users can specify a reasonable acquisition function to determine locations of next measurements[9,10].

[1]Jülich Centre for Neutron Science (JCNS) at Heinz Maier-Leibnitz Zentrum (MLZ), Forschungszentrum Jülich, Garching, Germany. [2]Laboratory for Neutron Scattering and Imaging, Paul Scherrer Institute (PSI), Villigen, Switzerland. ✉e-mail: m.teixeira.parente@fz-juelich.de; m.ganeva@fz-juelich.de

It was originally demonstrated on small-angle X-ray scattering (SAXS) and grazing-incidence small-angle X-ray scattering (GISAXS) applications. However, it was also applied to a TAS setting recently[11]. In reflectometry, experiments can be optimized by placing measurements at locations of maximum information gain using Fisher information[12,13]. Furthermore, the maximum a posteriori (MAP) estimator of a quantity of interest in a Bayesian setting[14,15] can be used to accelerate small-angle neutron scattering (SANS) experiments[16]. Moreover, neutron diffraction experiments are shown to benefit from active learning by incorporating prior scientific knowledge of neutron scattering and magnetic physics into GPR for an on-the-fly interpolation and extrapolation[17]. Materials synthesis and materials discovery were also considered as potential fields of application[18,19]. Interestingly, log-Gaussian (Cox) processes[20], the technique that we use as a basis for our approach, are applied in various domains such as epidemiology[21], insurance[22], geostatistics[23], or forestry[24,25]. The common factor of all those applications is the possibility to model or interpret their corresponding quantity of interest as an intensity.

From the field of artificial intelligence and machine learning, *active learning*[26–28] (also called *optimal experimental design* in statistics literature) provides a general approach that can be taken into account for the task of autonomous data acquisition in TAS experiments. In our context, an active learning approach sequentially collects intensity data while deciding autonomously where to place the next measurement. In other words, it updates its own source of information that its further decisions are based on.

The approach that we describe in this work regards an intensity distribution as a non-negative function over the domain of investigation and approximates it probabilistically by the mean function of a stochastic process. The primary objective for our choice of a stochastic process is that its posterior approximation uncertainties, i.e., its standard deviations after incorporating collected data, are largest in regions of signal as this enables us to identify them directly by maximizing the uncertainty as an acquisition function. More concretely, we fit a log-Gaussian process to the intensity observations, i.e., we apply *Gaussian process regression* (GPR)[29] to logarithmic intensities and exponentiate the resulting posterior process. The fact that GPR is a Bayesian technique that uses pointwise normal distributions to fit noisy data from an underlying inaccessible function of interest makes it indeed an interesting candidate for many applications and use cases where it is important to quantify some sort of uncertainty. However, we will see that, in our case, it is the logarithmic transformation of observed intensities that leads to large uncertainties in regions of signal and is thus the central element of our approach. Nevertheless, this approach is not only able to detect regions with strong signals, but also those with weak signals. A threshold parameter for intensity values and a background level, both estimated by statistics of an initial set of intensity observations on a particular grid, can control which signals are subject to be detected or neglected. Furthermore, costs for changing the measurement location caused by moving the axes of the instrument are respected as well.

Since gpCAM is the only of the mentioned approaches that was already applied to TAS[11], it is possible to briefly contrast it with ours. For the TAS setting, we see two major differences. First, gpCAM approximates the original intensity function (instead of its logarithm) which violates a formal assumption of GPR. In fact, GPR assumes that the function of interest is a realization of a Gaussian process. Since normal distributions have support on the entire real line, their realizations can, with positive probability, take negative values which is not possible for non-negative intensity functions. This issue does not occur in our methodology because we approximate the logarithm of the intensity function with GPR. Secondly, for identifying regions of signal, gpCAM requires users to specify an acquisition function based on a GPR approximation which can be problematic. Indeed, especially at the beginning of an experiment, having no or not much information about the intensity distribution, it can be difficult to find a reasonable acquisition function which leads to an efficient experiment. Moreover, even if users were successful in finding an acquisition function that works for a particular experiment, there is no guarantee that it is also applicable for another experiment with a different setting. Additionally, the acquisition function used in the only TAS experiment of gpCAM so far [ref. 11, Eq. (6)] makes it risk losing some of its interpretability[30,31] in the TAS setting, because physical units do not match and it is not obvious from a physics point of view why this function in particular is suitable for the goal of placing measurement points in regions of signal. In contrast, we are able to choose a particular canonical acquisition function that remains the same for each experiment since we identify regions of signal by the methodology itself or, more concretely, by the logarithmic transformation of a Gaussian process, and not primarily by an acquisition function as the critical component. Furthermore, the mentioned parameters of our approach, the intensity threshold and the background level, are both scalar values with, and not functions without a physical meaning. They are thus directly interpretable and can be estimated using initial measurements in a provably robust manner or set manually.

In this work, we demonstrate the applicability and benefits of our approach in two different ways. First, we present outcomes of a real neutron experiment performed at the thermal TAS EIGER[32] at the continuous spallation source SINQ of Paul Scherrer Institute (PSI) in Villigen, Switzerland. In particular, we compare with a grid approach and investigate the robustness of the results w.r.t. changes in the estimated background level and intensity threshold. It can be seen that our approach robustly identifies regions of signal, even those of small shape, and hence is able to improve the efficiency of this experiment. Moreover, we challenge our approach with a difficult initial setting and can demonstrate that its behaviour remains reasonable. Secondly, we apply a benchmark with several synthetic intensity functions and make fair comparisons with two competing approaches: an approach that places measurements uniformly at random and again a grid-based approach. In this setting, efficiency is measured by the time to reduce a relative weighted error in approximating the target intensity functions. The results show that our approach significantly improves efficiency for most intensity functions compared to the random and grid approach and is at least as efficient for the remainder. In addition, we can show that the results of our approach are robust w.r.t. changes in the intensity threshold parameter. Finally, we provide a comment to a comparison with gpCAM in this benchmark setting and a corresponding reference to the Supplementary Information.

## Results

### Problem formulation

From a methodological perspective, we aim to discover an intensity function $i : \mathcal{X} \to [0, \infty)$ on a rectangular set $\mathcal{X} \subseteq \mathbf{R}^n$, $n \in \mathbf{N}$, with coordinates on a certain hyperplane in four-dimensional **Q**-*E* space[8]. As an example, Fig. 1a displays an intensity function defined on $\mathcal{X} = [2.3, 3.3] \times [2.5, 5.5] \subseteq \mathbf{R}^2$, where the two-dimensional hyperplane is spanned by the vector $(0, 0, 1)$ in **Q** space with offset $(1, 1, 0)$ and energy transfer (*E*). For directions in **Q** space, we use relative lattice units (r.l.u.), while energy transfer *E* is measured in milli-electron volts (meV). Note that, due to restrictions of an instrument, the intensity function might only be defined on a subset $\mathcal{X}^* \subseteq \mathcal{X}$ consisting of all measurement locations reachable. For more details on the TAS setting, we refer to Supplementary Note 1.

The intensity function *i* is accessed by counting scattered neutrons on a detector device for a finite number of measurement locations $\mathbf{x} \in \mathcal{X}^*$ yielding noisy observations $I^*(\mathbf{x}) \sim \text{Pois}(\lambda = i(\mathbf{x}))$. Note that detector counts are usually normalized by neutron counts on a monitor device, i.e., the corresponding unit of measurement is detector counts/($M$ monitor counts), $M \in \mathbf{N}$. Since Poisson distributions $\text{Pois}(\lambda)$ with a sufficiently large parameter $\lambda > 0$ can be

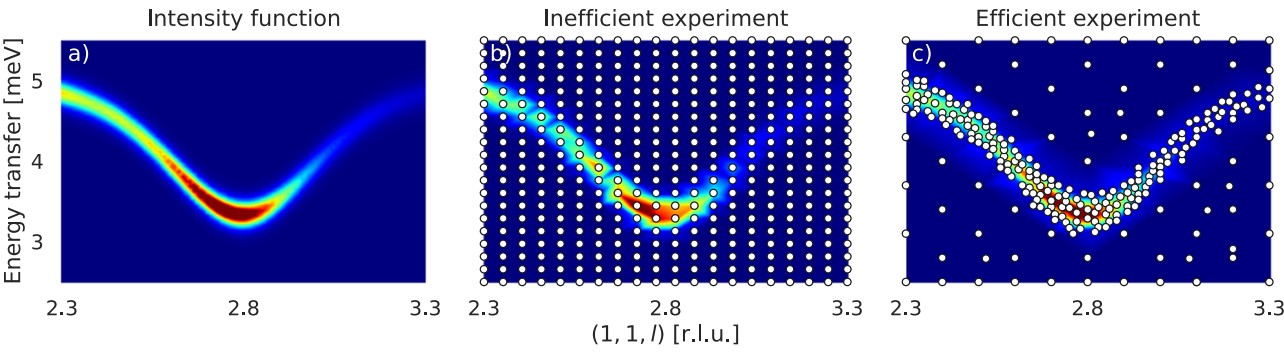

**Fig. 1 | Example of an intensity function and a corresponding inefficient and efficient experiment.** For the **Q** direction, we use relative lattice units (r.l.u.). **a** Intensity function on $\mathcal{X} = [2.3, 3.3] \times [2.5, 5.5]$ along the **Q** direction $(0, 0, 1)$ with offset $(1, 1, 0)$ and energy transfer. The color spectrum ranges from blue (low signal) to red (high signal). **b** Inefficient experiment with a large part of measurement locations (dots) in the background (dark blue area). **c** More efficient experiment with most measurement locations in the region of signal.

---

## BOX 1

# General experiment algorithm

---

**Data:** cost measure $c$, cost budget $C \geq 0$

1 $\mathcal{A} \leftarrow ()$
2 $J = 0$
3 **while** $c(\mathcal{A}) < C$ **do**
4 Determine next measurement location $\mathbf{x}_{J+1} \in \mathcal{X}^*$
5 Observe noisy intensity $\hat{i}_{J+1}$ at $\mathbf{x}_{J+1}$
6 $\mathcal{A} \leftarrow ((\mathbf{x}_1, \hat{i}_1), \ldots, (\mathbf{x}_J, \hat{i}_J), (\mathbf{x}_{J+1}, \hat{i}_{J+1}))$
7 $J \leftarrow J + 1$
8 **end**

---

approximated by a normal distribution $\mathcal{N}(\lambda, \lambda)$, we assume that

$$I^+(\mathbf{x}) = i(\mathbf{x}) + \sqrt{i(\mathbf{x})}\eta^+, \tag{1}$$

where $\eta^+ \sim \mathcal{N}(0, 1)$.

In the following, we repeatedly refer to "experiments" which are defined as a sequential collection of intensity observations.

**Definition 1.** (Experiment). An *experiment* $\mathcal{A}$ is an $N$-tuple of location-intensity pairs

$$\mathcal{A} = ((\mathbf{x}_1, \hat{i}_1), \ldots, (\mathbf{x}_N, \hat{i}_N)) \tag{2}$$

where $|\mathcal{A}| := N \in \mathbf{N}$ denotes the number of measurement points, $\mathbf{x}_j \in \mathcal{X}^*$ are measurement locations, and $\hat{i}_j$ are corresponding noisy observations of an intensity function $i$ at $\mathbf{x}_j$.

Since demand for beam time at TAS is high but availability is limited, the goal of an approach is to perform the most informative experiments at the lowest possible cost. Beam time use can, for example, be optimized when excessive counting in uninformative regions like the background (Fig. 1b) is avoided but focused on regions of signal (Fig. 1c).

We quantify the benefit of an experiment $\mathcal{A}$ by a benefit measure $\mu = \mu(\mathcal{A}) \in \mathbf{R}$ and its cost by a cost measure $c = c(\mathcal{A}) \in \mathbf{R}$. For now, it suffices to mention that cost is measured by experimental time (the time used for an experiment) and benefit is defined by reducing a relative weighted error between the target intensity function and a

corresponding approximation constructed by the collected intensity observations. Note that, quantifying benefits this way, their computation is only possible in a synthetic setting with known intensity functions (as in our benchmark setting). Real neutron experiments do not meet this requirement and thus must be evaluated in a more qualitative way.

In our setting, an approach attempts to conduct an experiment $\mathcal{A}$ with highest possible benefit using a given cost budget $C \geq 0$, i.e., it aims to maximize $\mu(\mathcal{A})$ while ensuring that $c(\mathcal{A}) \leq C$. The steps of a corresponding general experiment are given in Box 1. Line 4 is most important and crucial for both cost and benefit of the experiment since it decides where to observe intensities, i.e., count neutrons, next.

From an algorithmic perspective, if we denote the current step of an experiment by $J \in \mathbf{N}$, our approach implements the decision for the next measurement location $\mathbf{x}_{J+1} \in \mathcal{X}^*$ by maximizing an objective function $\phi_J : \mathcal{X}^* \to \mathbf{R}$. It balances an acquisition function $\text{acq}_J : \mathcal{X}^* \to \mathbf{R}$ and a cost function $c_J : \mathcal{X}^* \to \mathbf{R}$ that both depend on the current step $J$ and hence can change from step to step. The acquisition function $\text{acq}_J$ indicates the value of any $\mathbf{x} \in \mathcal{X}^*$ for improving the benefit of the experiment whereas the cost function $c_J$ quantifies the costs of moving the instrument axes from the current location $\mathbf{x}_J \in \mathcal{X}^*$ to $\mathbf{x}$. The metric $d : \mathcal{X}^* \times \mathcal{X}^* \to [0, \infty)$ used for our particular cost function

$$c_J(\mathbf{x}) := d(\mathbf{x}_J, \mathbf{x}) \tag{3}$$

is formally specified in Supplementary Note 1 (Eq. (11)).

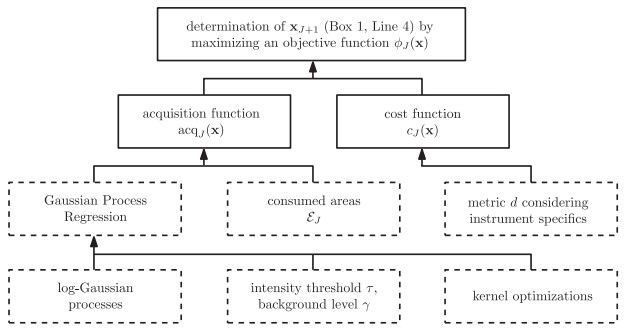

**Fig. 2 | Schematic representation of our approach in the context of the general experiment algorithm.** The general components (solid line boxes) are composed of the specific components (dashed line boxes) which mainly form our methodology.

Our methodology concentrates on developing a useful acquisition function as the crucial component of our approach making it straightforward to find regions of signal. A schematic representation with general and specific components of our approach in the context of the general experiment algorithm (Box 1) can be found in Fig. 2. Specific components necessary to understand the experimental results are informally introduced below. Details for these specific components as well as our final algorithm (Box 2), concretizing the general experiment algorithm, along with all its parameters and their particular values used are specified in the Methods section.

### Log-Gaussian processes for TAS

We briefly describe here why log-Gaussian processes, our central methodological component, are suitable to identify regions of signal in a TAS experiment and thus used to specify a reasonable acquisition function acq$_J$. Methodological details can be found in the Methods section.

Although the intensity function is not directly observable due to measurement noise (Eq. (1)), we aim to approximate it by the mean function of a log-Gaussian process

$$I(\mathbf{x}) := \exp(F(\mathbf{x})), \qquad (4)$$

where $F$ is a Gaussian process. That is, after $J$ steps, we fit logarithmic intensity observations to $F$ yielding its posterior mean and variance function denoted by $m_J$ and $\sigma_J^2$, respectively. The acquisition function is then defined as the uncertainty of $I$ given by its posterior standard deviation, i.e.,

$$\mathrm{acq}_J(\mathbf{x}) = \sqrt{(\exp(\sigma_J^2(\mathbf{x})) - 1) \cdot (\exp(2m_J(\mathbf{x}) + \sigma_J^2(\mathbf{x})))}. \qquad (5)$$

Observe the crucial detail that $m_J$ appears exponentially in this function which is the main reason why our approach is based on log-Gaussian processes. A posterior log-Gaussian process is thus able to find regions of signal just through maximizing its uncertainty. As illustration, regard Fig. 3 displaying the posterior of the Gaussian process $F$ together with logarithmic intensity observations (Fig. 3a) and the corresponding posterior of the log-Gaussian process $I$ (Fig. 3b).

### Intensity threshold and background level

Maximizing the acquisition function from Eq. (5) prioritizes regions with high intensities over regions with low intensities. This poses a problem when there are multiple signal regions with intensities of different magnitudes (Supplementary Fig. 1a). Indeed, measurement points are mainly placed in regions with higher intensities whereas regions with less signal are neglected (Supplementary Fig. 1b). In TAS, we are interested in each region of signal no matter of which intensity magnitude. We compensate for this potential problem by introducing

an intensity threshold $\tau > 0$ for observed intensities. That is, we truncate the observed intensities to a maximum of $\tau$ before fitting (Supplementary Fig. 1c). Consequently, measurement points get more evenly distributed among all signal regions (Supplementary Fig. 1d) since their placement is not biased due to large differences in their intensity values.

As another problem, intensity observations, in neutron experiments, contain background which is not part of the actual signal, i.e., even if there is no actual signal at a certain location, we might nonetheless observe a positive intensity there. If our approach does not compensate for regions of background, it might not recognize them as parasitic and hence consider them as regions of weak signal which potentially yields uninformative measurement points being placed there. Therefore, we subtract a background level $\gamma \in [0, \tau]$ from already threshold-adjusted intensity observations while ensuring a non-negative value.

In an actual experiment, both, the intensity threshold and the background level, are estimated by statistics of initial measurement points which is described in more detail in the Methods section. The estimation of the intensity threshold however depends on a parameter $\beta \in (0, 1]$ controlling the distinction between regions of strong and weak signals (Eq. (47)) that needs to be set before starting an experiment. This parameter is already mentioned here as we examine its impact on our results in the benchmark setting described below.

### Neutron experiment

At SINQ (PSI), we investigated a sample of SnTe (tin telluride) in a real neutron experiment performed at the thermal TAS EIGER[32]. Our general aim is to reproduce known results from [ref. [33], Fig. 1b] using our approach. Furthermore, we assess the robustness of the experimental result w.r.t. changes in the parameters for the background level and the intensity threshold (scenario 1) as well as challenge our approach using a coarser initialization grid on a modified domain $\mathcal{X}$ with no initial measurement locations directly lying in a region of signal (scenario 2).

The software implementation of our approach communicates with the instrument control system NICOS (nicos-controls.org) which was configured on site. For the experimental setting at the instrument, we refer to Supplementary Note 2.

As mentioned, the benefit measure used for benchmarking, involving a known target intensity function, is not computable in this setting of a neutron experiment due to experimental artefacts like background and noise. We, therefore, evaluate the results of our approach in a more qualitative way for this experiment. Also, although the costs for moving the instrument axes contribute to the total experimental time, we do not consider them here for optimizing the objective function since they are approximately constant across the domain $\mathcal{X}$.

For scenario 1, we adopt the setting from the original results [ref. [33], Fig. 1b] which, in our context, means to investigate intensities on $\mathcal{X} = [0, 2] \times [2, 12.4]$ along the vector $(1, 1, 0)$ in $\mathbf{Q}$ space with offset $(0, 0, 3)$ and energy transfer. Initially, we performed measurements in a conventional mode for reference, i.e., we mapped $\mathcal{X}$ with a grid of 11 columns containing 27 measurement points each (bottom row in Fig. 4). These measurements were arranged in columns from bottom to top, i.e., from low to high energies, and from left to right, i.e., from small to large values for the coordinate along the vector in $\mathbf{Q}$ space.

A direct comparison with our approach would put this mapping mode in disadvantage since its total experimental time could have been spent more efficiently. For this reason, the approach that we eventually take for comparison, the grid approach (top row in Fig. 4), takes the measurement points from the mapping mode but changes their order into four stages I-IV (Supplementary Fig. 2a–d). The first stage is from bottom to top and from left to right but only consists of every other grid point on both axes. The second stage is again from

## BOX 2

## Final algorithm

**Data:** odd number of rows $N_{\text{row}}$ for initialization grid, parameters $\Delta_{\max}^{\text{rel}}, \Delta_{\min}^{\text{abs}}, l_{\max}$, and $\beta$ for estimating the background level and intensity threshold, matrix-valued function $E(\mathbf{x})$ for elliptic consumed areas, objective function $\phi_J$, kernel $\kappa_\theta$ and optimizer for hyperparameters $\theta$ (Eq. (19)), stopping criterion for kernel optimizations $P_{\text{KO}} = P_{\text{KO}}(J)$, cost measure $c = c(\mathcal{A})$, cost budget $C \geq 0$

1  Observe noisy intensities $\hat{i}_1, \ldots, \hat{i}_{J_0}$ at initial locations $\mathbf{x}_1, \ldots, \mathbf{x}_{J_0}$
2  $\mathcal{A}_0 = ((\mathbf{x}_1, \hat{i}_1), \ldots, (\mathbf{x}_{J_0}, \hat{i}_{J_0}))$
3  Optimize hyperparameters $\theta_{J_0}$ of kernel $\kappa_{J_0}$
4  Compute background level $\gamma = \gamma(\mathcal{A}_0)$ and intensity threshold $\tau = \tau(\mathcal{A}_0) > \gamma$
5  $\mathcal{A} = \mathcal{A}_0$
6  $J = J_0$
7  **while** $c(\mathcal{A}) < C$ **do**
8 Determine next measurement location $\mathbf{x}_{J+1} := \arg\max_{\mathbf{x} \in \mathcal{X}^*} \phi_J(\mathbf{x})$
9 Observe noisy intensity $\hat{i}_{J+1}$ at $\mathbf{x}_{J+1}$
10 **if** *kernel optimizations stopped* **then**
11 $\theta_{J+1} = \theta_J$
12 **else**
13 Optimize hyperparameters $\theta_{J+1}$ of kernel $\kappa_{J+1}$
14 **if** $P_{\text{KO}}(J+1)$ **then**
15 Stop kernel optimizations
16 **end**
17 **end**
18 $\mathcal{A} \leftarrow ((\mathbf{x}_1, \hat{i}_1), \ldots, (\mathbf{x}_J, \hat{i}_J), (\mathbf{x}_{J+1}, \hat{i}_{J+1}))$
19 $J \leftarrow J + 1$
20 **end**

bottom to top but from right to left and also consists only of every other grid point on both axes. The third and fourth stage are analogous but consist of the remaining grid points that were skipped in the first two stages. With this order, the grid approach can observe intensities on each region of $\mathcal{X}^*$ already after the first stage and hence can acquire more information in a faster way as with the conventional order.

We ran our approach in the default setting (Table 1), i.e., with automatically computed background level and intensity threshold, as well as with two alternative pairs of corresponding parameter values manually set in order to study the robustness of the results. In the default setting (row 2 in Fig. 4), the background level and intensity threshold were computed to $\gamma_0 = 58$ and $\tau_0 = 94.5$, respectively. To study the robustness of these results w.r.t. changes in the intensity threshold, the parameters for the first alternative (row 3 in Fig. 4) were set to $\gamma_1 = \gamma_0$ and $\tau_1 = 130$. The second alternative (row 4 in Fig. 4), for studying the robustness w.r.t. changes in the background level, was configured with $\gamma_2 = 30$ and $\tau_2 = \tau_0$. Eventually, the cost budget $C$ for each instance of our approach was determined by the total experimental time of the grid approach which was ~11.5 hours. Each respective initialization with the same grid of 61 measurement points took ~2.9 h of experimental time. The results displayed in Fig. 4 show that, after initialization, our approach places measurement points mainly in regions of signal in all three settings. The grid approach, in contrast, has observed intensities at a considerable amount of uninformative measurement locations. Note that the evolution of the experiments related to the three different settings of our approach can be seen in Supplementary Movies 1-3.

In scenario 2, we tried to challenge our approach with a more difficult initial setting. The initialization grid in scenario 1 (triangles in Fig. 4) indeed beneficially lies on an axis of symmetry of the intensity function. Also, partly as a consequence, some initial measurements points are already located in regions of signal. From a methodological perspective, it is interesting to investigate how our approach behaves after unfavorable initialization, i.e., if almost all initial intensity observations are lying in the background and hence almost no useful initial information is available. Hence, we reduced the number of initial measurement points from 61 to 25 and modified the domain to $\mathcal{X} = [0.2, 2] \times [2, 16]$ to break the axis of symmetry and measure larger energy transfers (Fig. 5a). Estimating the background level and intensity threshold is not reasonable in this setting since the amount of initial information is too little. Therefore, we manually set $\gamma = 50$ and $\tau = 80$, respectively. The cost budget $C$ for our approach was not set explicitly here. In fact, we stopped the experiment manually after ~7.3 h of total experimental time, with ~1.1 h spent on initialization with the modified grid. The result is depicted in Fig. 5b. It shows that, even in this challenging situation, our approach keeps making reasonable decisions on measurement locations. The evolution of the experiment related to this scenario can be seen in Supplementary Movie 4.

### Benchmark

This section shows results of a benchmark on several two-dimensional (i.e., $n = 2$) test case intensity functions (Supplementary Fig. 3) as a quantitative proof of concept. The benchmarking procedure quantifies the performance of an approach by how much benefit it is able to

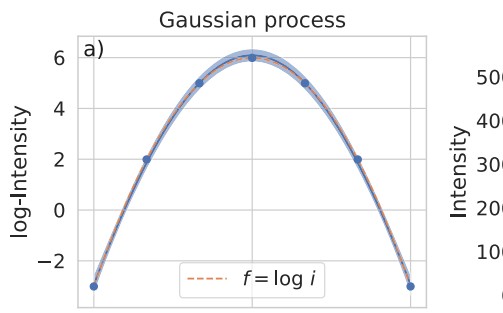

**Fig. 3 | Transformation of a Gaussian process to a corresponding log-Gaussian process.** The one-dimensional example intensity function $i$ and its logarithm are displayed with dashed orange lines. **a** Gaussian process with observations (blue dots) of $\log i$ and its uncertainties (light blue area) around its mean function (solid blue line). **b** Corresponding log-Gaussian process. As expected from Eq. (5), maximizing the uncertainty of the log-Gaussian process enables to find regions of signal.

**Fig. 4 | Results for scenario 1.** For the **Q** direction, we use relative lattice units (r.l.u.). Each row displays the results of a certain approach. The columns indicate the four stages (I-IV) of the grid approach in the top row (**a**–**d**). Rows 2-4 correspond to results of our approach in three different settings. Triangles represent the initialization grid and dots show locations of intensity observations autonomously placed by our approach. Row 2 (**e**–**h**) corresponds to the default setting ($\gamma_0 = 58$ and $\tau_0 = 94.5$ were computed automatically). Manually changing the intensity threshold to $\tau_1 = 130$ (with $\gamma_1 = \gamma_0$) leads to results depicted in row 3 (**i**–**l**). Row 4 (**m**–**p**) shows results after changing the background level to $\gamma_2 = 30$ (with $\tau_2 = \tau_0$). The bottom row (**q**–**t**) shows the mapping in its conventional order for completeness.

**Table 1 | Default parameter values used for the final algorithm**

| Parameter | $N_{row}$ | $\Delta_{max}^{rel}$ | $\Delta_{min}^{abs}$ | $l_{max}$ | $\beta$ | $N_{\mathcal{H}}$ | $\mathcal{H}$ | $N_{KO}^{min}$ | $N_{KO}^{max}$ | $k_{KO}$ | $\varepsilon_{KO}$ | $\delta^-$ | $\delta^+$ |
|---|---|---|---|---|---|---|---|---|---|---|---|---|---|
| Value | 11 | 0.5 | 15 | 6 | 0.5 | 100 | $[10^{-3}, 10^2]^3$ | 25 | 75 | 9 | 0.025 | $10^{-3}$ | 1 |

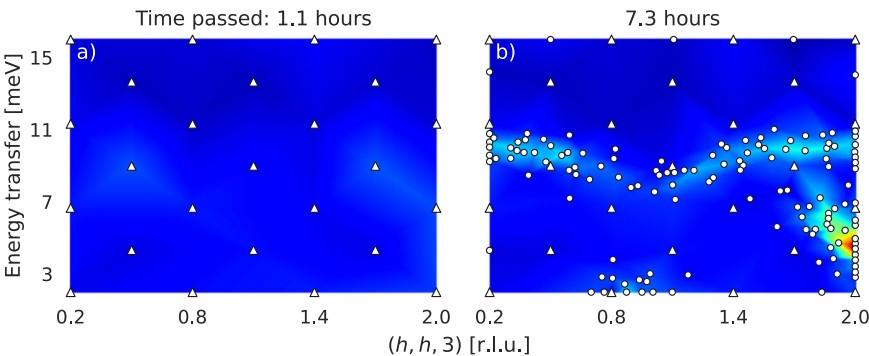

**Fig. 5 | Results for scenario 2.** For the **Q** direction, we use relative lattice units (r.l.u.). **a** Reduced number of initial measurement points (triangles) in a modified domain providing almost no initial information about the intensity function. **b** Intensity observations (dots) autonomously placed after the uninformative initialization.

acquire for certain cost budgets in the context of predefined test cases. We briefly describe its setting in the following paragraph. For more details, we refer to the original work[34].

A test case mainly includes an intensity function defined on a certain set $\mathcal{X}$ and a synthetic TAS used for moving on $\mathcal{X}^*$ with certain velocities and observing intensities. As mentioned, the cost measure is chosen to be the experimental time used, i.e., the sum of cumulative counting time and cumulative time for moving instrument axes. The benefit measure is defined by a relative weighted $L^2$ approximation error between the target intensity function and a linear interpolation $\hat{i} = \hat{i}(\mathcal{A})$ using observed intensities from experiments $\mathcal{A}$. It encodes the fact that a TAS experimenter is more interested in regions of signal than in the background which suggests to use $i$ itself as a weighting function. However, an important constraint, which is controlled by a benchmark intensity threshold $\tau^* > 0$ truncating weights to

$$i_{\tau^*}(\mathbf{x}) := \min\{i(\mathbf{x}), \tau^*\}, \qquad (6)$$

as similarly described for the intensity threshold of our approach, is that separate regions of signal with different magnitudes of intensity might be equally interesting. Formally, we define

$$\mu(\mathcal{A}) := \frac{\|i - \hat{i}(\mathcal{A})\|}{\|i\|}, \qquad (7)$$

where $\| \cdot \| = \| \cdot \|_{L^2(\mathcal{X}^*, \rho_{i,\tau^*})}$ for the weighting function

$$\rho_{i,\tau^*}(\mathbf{x}) := \frac{i_{\tau^*}(\mathbf{x})}{\int_{\mathcal{X}^*} i_{\tau^*}(\mathbf{x}') \, d\mathbf{x}'}. \qquad (8)$$

For each test case, benefits are measured for several ascending cost budgets, called "milestone values", to demonstrate the evolution of performance over time. We note that this synthetic setting includes neither background nor measurement noise as both are artefacts of real neutron experiments.

We run the benchmarking procedure with three approaches for comparison:
1. random approach,
2. grid approach,
3. our approach with different values for the intensity threshold parameter $\beta$.

The random approach places intensity observations uniformly at random in $\mathcal{X}^*$. The grid approach is adopted from the section on the neutron experiment but using a square grid of dimension $P \times P$, $P \in \mathbf{N}$.

For our approach, we set a zero background level, i.e., $\gamma = 0$, manually since background is not included in this synthetic setting. Our approach is run with four variations of the intensity threshold parameter $\beta$ (Eq. (47)) in order to study corresponding sensitivities of the benchmark results.

The specific benchmarking procedure involves four milestone values according to the four stages (I-IV) of the grid approach, i.e., the $m$-th milestone value represents the experimental time needed to complete stage $m \in \{I, II, III, IV\}$. Observe that they depend on the particular test case. The specific milestone values used are indicated in Supplementary Note 3. The number of columns/rows $P$ for the grid approach is chosen to be the maximum number such that the corresponding experimental time for performing an experiment in the described order does not exceed 9 hours.

Since both the random and our approach contain stochastic elements, we perform a number of 100 repeated runs with different random seeds for each test case in order to see the variability of their results.

The results for each test case along with its corresponding intensity function are shown in Fig. 6. Our approach has, on median average, performed significantly better than the random and grid approach in most test cases. Furthermore, its mostly thin and congruent shapes of variability (light color areas) demonstrate its reproducibility and its robustness w.r.t. changes in the intensity threshold parameter $\beta$. Examples of particular experiments performed by our approach are additionally provided in Supplementary Fig. 4 for each test case.

A well-formulated and sustainable comparison of our approach with gpCAM in this benchmark setting is currently difficult to implement because, to the best of our knowledge, gpCAM does not specify how to choose its main parameter, the acquisition function, in the TAS setting. In this situation, where relevant information is missing, it would be inappropriate to compare the two approaches at this point. We have therefore decided to provide a comparison that is currently possible, i.e., based on all the information available to us about gpCAM, in Supplementary Note 4. The acquisition function used there is chosen to be the same as that used for the neutron experiment at the TAS ThALES (ILL) [ref. 11, Eq. (6)].

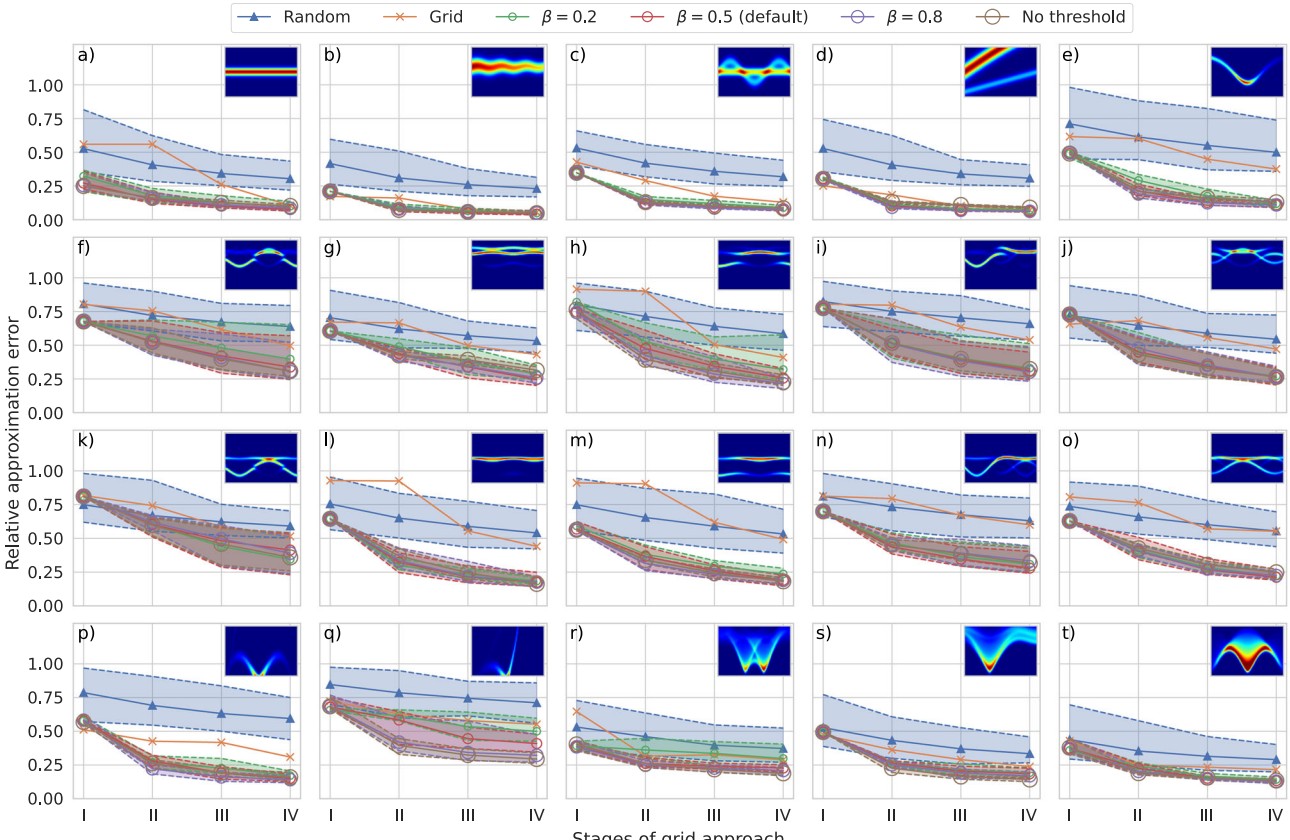

**Fig. 6 | Benchmark results.** For each approach, every subfigure (**a**–**t**) plots the decay of a relative approximation error (Eq. (7)) between the target intensity function of the corresponding test case (top right corners) and a linear interpolation of collected intensity observations for four milestone values (symbols) which are determined by the four stages (I-IV) of the grid approach. The solid lines show medians of the resulting benefit values, whereas the light color areas with dashed boundaries indicate the range between their minimum and maximum to visualize their variability caused by stochastic elements.

## Discussion

The results of the neutron experiment demonstrate the benefits of our approach. Indeed, in scenario 1 (Fig. 4), our approach identifies regions of strong as well as of weak signal in each setting and even finds isolated relevant signals of small shape at the edge repeatedly. Therefore, for this experimental setting, the results are shown to be robust w.r.t. changes in the estimated background level and intensity threshold, which we view as an important outcome of this experiment. The choice of these parameters is however directly reflected in the placement of measurement points which indicates a certain aspect of interpretability and explainability[30,31] of our approach. An intensity threshold higher in value namely leads to measurement points that are placed on a thinner branch of signal (row 3 in Fig. 4), whereas as a lower background level yields more exploratory behaviour, with a risk to measure in regions of background from time to time (row 4 in Fig. 4). Additionally, note that a smaller intensity threshold results in measurement points also being placed in regions of high intensity gradient.

Furthermore, in each setting, our approach (rows 2–4 in Fig. 4) has significantly fewer measurement points in the background compared to the grid approach (top row in Fig. 4) as expected and thus uses experimental time more efficiently. The grid approach additionally does not cover small signal regions at the edge. A simulated intensity between the two regions of signal on the right [ref. 33, Fig. 1c] is not seen by our approach which is, however, in agreement with the original experiment [ref. 33, Fig. 1b]. In situations like this, a human experimenter can focus on such details and place additional measurement points manually, if necessary.

When applying GPR, we use a common Gaussian kernel as detailed in the Methods section. The (logarithm of the) intensity function from

Fig. 4 however violates the stationarity of this kernel. Indeed, using a stationary kernel assumes homogeneous statistical behaviour of the intensity function across the entire domain $\mathcal{X}$ which is not the case for our particular scenario. The length scale along the $E$ axis, for instance, is differing for different values on the **Q** axis. In the middle of the **Q** axis, the length scale is certainly larger than near its edges. Note that the length scale along the **Q** axis is also non-stationary. These discrepancies are one of the main reasons for our choice of the material SnTe and the setting mentioned above since it provides an opportunity to demonstrate that a stationary kernel, which is computationally substantially cheaper than non-stationary kernels, is sufficient for identifying regions of signal and hence for performing efficient experiments.

In scenario 2 (Fig. 5), we challenged our approach with a difficult setting. Although the initialization grid and the domain were organized such that no initial measurement point is located in a region of signal, and hence the approach is initialized with little useful information, its behaviour stays reasonable. It namely keeps placing measurements in regions of signal and can still identify the small region with strong signal on the bottom right. However, the signal region in the middle of the domain is not fully identified, which can be explained by the relatively short experimental time (7.3 hours) as well as by the stationarity of the kernel since the sparse data suggest a short length scale along the $E$ axis leading to the assumption of lower function values in an actual region of signal.

Note that the reduced amount of initial measurement points in scenario 2 is only applied for the purpose of challenging our approach and a demonstration of its robustness. In productive operation, however, we stay with the larger initialization grid from scenario 1 since

placing some measurements in regions of background for a proper determination of the same is a valuable step during an experiment providing relevant information for the experimenter. Fortunately, this mostly leads to a sufficiently good initialization of our approach, despite using a common stationary kernel.

For further results of neutron experiments in the setting from scenario 1 as well as from an additional scenario (scenario 3) that investigated another phonon of SnTe in the default setting of our approach, we refer to Supplementary Note 5.

The benchmark results (Fig. 6), as a proof of concept, confirm the outcomes of the neutron experiment quantitatively in a synthetic setting. Our approach shows a better performance, measured by a relative weighted approximation error (Eq. (7)) between a target intensity function and a linear interpolation of collected intensity observations, compared to the random and grid approach for all test cases. The improvements are especially substantial for target intensity functions with smaller regions of signal (Fig. 6a–q). For intensity functions that cover a substantial part of the domain $\mathcal{X}$ (Fig. 6r–t), the competing approaches are also able to place intensity observations in regions of signal early during the experiment and hence it is difficult for any approach to demonstrate its benefit in these scenarios.

The variability in the results of our approach containing stochastic elements, quantified by the median of benefit values and the range between their minimum and maximum, is shown to be small for most test cases and acceptable for the remainder. Our approach can thus be considered reliable with reproducible results despite starting with a different sequence of pseudo-random numbers for each run.

Moreover, the benchmark results indicate that our approach is robust w.r.t. changes in the intensity threshold. The four variations of the corresponding controlling parameter $\beta$ (Eq. (47)) yield similar results for most test cases. It should be noted, however, that the use of a reasonable value for $\beta$ in real neutron experiments is nevertheless important. Indeed, it not only eliminates the effect of outliers in initial intensities but also, recall, allows to control the width of signal branches on which the measurement points are placed, and thus the extent to which regions of high gradient are preferred over those of peak intensity.

In addition to the results of the neutron experiment, the particular experiments performed by our approach in the benchmark setting (Supplementary Fig. 4) confirm that it, after initialization, autonomously places a large part of measurement points in regions of signal for a variety of different intensity distributions.

As a conclusion, in the previous sections, we demonstrated that our approach indeed improves the efficiency of TAS experiments and allows to make better use of the experimenter's time. It maximizes an acquisition function given by approximation uncertainties of a log-Gaussian process in order to find informative measurement points and not waste experimental time in regions such as the background. Our robust and reproducible results suggest that it is in fact capable of autonomously obtaining a rapid overview of intensity distributions in various settings.

In a real neutron experiment at the thermal TAS EIGER, our approach repeatedly demonstrated its ability to identify regions of signal successfully leading to a more efficient use of beam time at a neutron source of a large-scale research facility. It was additionally shown that it keeps making reasonable decisions even when being initialized with little information. Furthermore, substantial performance improvements, in comparison with two competing approaches, and their robustness were quantified in a synthetic benchmark setting for numerous test cases.

Nevertheless, we feel that the automated estimation of algorithmic parameters (background level and intensity threshold) from statistics of initial measurements, despite its good performance in each of the settings discussed, needs to be confirmed in future experiments.

Our approach, so far, solves the problem of where to measure. However, an interesting topic of future research is the major question of how to measure at a certain location. Counting times were assumed to be constant in this work and thus promise to be another possibility to save experimental time if determined autonomously in an advantageous way. Indeed, large counting times in regions of high intensities and background are actually not needed since the necessary information can also be collected in less time. Experimenters, in contrast, are often more interested in weaker signals and their comprehensive measurement to reduce corresponding error bars.

Moreover, although using a common stationary kernel for GPR has proven to be sufficient for identifying regions of signal, we regard the application of non-stationary kernels with input-dependent hyperparameters[35–38], e.g., length scales, also modelled by log-Gaussian processes, as another interesting option for further investigations.

Finally, a certainly more involved topic for future research is the parameter estimation for physical models such as Hamiltonians, if available, in a Bayesian framework using data collected by an autonomous approach. Once this estimation is possible, a natural follow-up question is how our approach, which is completely model-agnostic, i.e., it does not take into account any information from a physical model, compares to a model-aware approach in terms of reducing uncertainties of parameters to be estimated while minimizing experimental time.

## Methods
Our approach is methodologically based on *Gaussian process regression* (GPR)[29], a Bayesian technique for estimating and approximating functions from pointwise data that allows to quantify uncertainties on the approximation itself in the form of normal distributions. We fit a Gaussian process to logarithmic intensity observations and exponentiate the resulting posterior process yielding a log-Gaussian process. As mentioned, this transformation causes approximation uncertainties to be higher in regions of signal which in turn can be exploited for the definition of a useful acquisition function in the TAS setting.

### Gaussian process regression
We generally intend to approximate a function of interest $f : \mathcal{X} \to \mathbf{R}$, which becomes the logarithm of the intensity function later. Using GPR for this, we have to assume that $f$ is a realization of a Gaussian process $F$.

**Definition 2.** (Gaussian process). A *Gaussian process*

$$F \sim \mathcal{GP}(m(\mathbf{x}), \kappa_\theta(\mathbf{x}, \mathbf{x}')) \tag{9}$$

with (prior) mean function $m : \mathcal{X} \to \mathbf{R}$ and parameterized kernel function $\kappa_\theta : \mathcal{X} \times \mathcal{X} \to [0, \infty)$ is a collection of random variables $(F(\mathbf{x}))_{\mathbf{x} \in \mathcal{X}}$ such that for any finite amount of evaluation points $\mathbf{x}^{(\ell)} \in \mathcal{X}$, $\ell = 1, \ldots, L, L \in \mathbf{N}$, the random variables $F^{(\ell)} := F(\mathbf{x}^{(\ell)})$ follow a joint normal distribution, i.e.,

$$(F^{(1)}, \ldots, F^{(L)})^\top \sim \mathcal{N}(\mathbf{m}, \mathcal{K}), \tag{10}$$

where

$$\mathbf{m} = (m(\mathbf{x}^{(1)}), \ldots, m(\mathbf{x}^{(L)}))^\top \in \mathbf{R}^L \quad \text{and} \quad \mathcal{K}_{\ell_1, \ell_2} = \kappa_\theta(\mathbf{x}^{(\ell_1)}, \mathbf{x}^{(\ell_2)}) \geq 0. \tag{11}$$

Note that, for each $\mathbf{x} \in \mathcal{X}$, it particularly holds that

$$F(\mathbf{x}) \sim \mathcal{N}(m(\mathbf{x}), \sigma^2(\mathbf{x})), \tag{12}$$

where $\sigma^2(\mathbf{x}) := \mathbf{Var}(F(\mathbf{x})) = \kappa_\theta(\mathbf{x}, \mathbf{x}) \geq 0$ is the (prior) variance function.

The kernel function $\kappa_\theta$ is an important component in GPR since it describes the correlation structure between random variables $F(\mathbf{x})$. Hence, it enables to include assumptions on realizations of $F$ and determines function properties like regularity, periodicity, symmetry, etc. In practice, it is crucial to choose a kernel function that matches with properties of the function of interest $f$.

We acquire knowledge on $f$ through noisy observations $\hat{f}_j$ at locations $\mathbf{x}_j \in \mathcal{X}^*$ (Eq. (2)) in our context. Therefore, for $j = 1, \ldots, J$, we set

$$\hat{F}(\mathbf{x}_j) = F(\mathbf{x}_j) + e(\mathbf{x}_j)\eta_j, \tag{13}$$

where $\eta_j \sim \mathcal{N}(0,1)$ are i.i.d. random variables independent of $F(\mathbf{x}_j)$ and $e(\mathbf{x}) > 0$ denotes the noise standard deviation at $\mathbf{x} \in \mathcal{X}$. Note that $\hat{F}(\mathbf{x}_j)$ is a normally distributed random variable.

For clear notation, we define

$$X_J := \begin{pmatrix} | & & | \\ \mathbf{x}_1 & \cdots & \mathbf{x}_J \\ | & & | \end{pmatrix} \in \mathbf{R}^{n \times J} \quad \text{and} \quad \hat{\mathbf{f}}_J := (\hat{f}_1, \ldots, \hat{f}_J)^\top \in \mathbf{R}^J, \tag{14}$$

and let

$$h(X_J) := (h(\mathbf{x}_1), \ldots, h(\mathbf{x}_J))^\top \in \mathbf{R}^J \tag{15}$$

for any function $h : \mathcal{X} \to \mathbf{R}$.

After observations have been made, we are interested in the posterior, i.e., conditional, Gaussian process. It holds that

$$F(\mathbf{x}) \mid (\hat{F}(X_J) = \hat{\mathbf{f}}_J) \sim \mathcal{N}(m_J(\mathbf{x}), \sigma_J^2(\mathbf{x})) \tag{16}$$

with the posterior mean function

$$m_J(\mathbf{x}) = m(\mathbf{x}) + \kappa_J(\mathbf{x}, X_J)^\top \left[ \kappa_J(X_J, X_J) + \text{diag}(e(X_J))^2 \right]^{-1} (\hat{\mathbf{f}}_J - m(X_J)) \tag{17}$$

and the posterior variance function

$$\sigma_J^2(\mathbf{x}) = \sigma^2(\mathbf{x}) - \kappa_J(\mathbf{x}, X_J)^\top \left[ \kappa_J(X_J, X_J) + \text{diag}(e(X_J))^2 \right]^{-1} \kappa_J(\mathbf{x}, X_J). \tag{18}$$

If necessary, the hyperparameters $\theta_J$ of the kernel function $\kappa_J := \kappa_{\theta_J}$ can be optimized using data $\hat{\mathbf{f}}_J$. For this, we compute $\theta_J$ such that the logarithm of the so-called marginal likelihood, i.e.,

$$\log \rho_{\hat{F}(X_J)}(\hat{\mathbf{f}}_J) = \log \left( \int_{\mathbf{R}^J} \rho_{\hat{F}(X_J)|F(X_J)}(\hat{\mathbf{f}}_J|\mathbf{f}) \, \rho_{F(X_J)}(\mathbf{f}) \, d\mathbf{f} \right)$$
$$= -\frac{1}{2}(\hat{\mathbf{f}}_J - m(X_J))^\top \left[ \kappa_J(X_J, X_J) + \text{diag}(e(X_J))^2 \right]^{-1} (\hat{\mathbf{f}}_J - m(X_J)) \tag{19}$$
$$- \frac{1}{2}\log|\kappa_J(X_J, X_J) + \text{diag}(e(X_J))^2| - \frac{n}{2}\log 2\pi,$$

is maximized. A suitable optimizer is specified below. Note that the analytical expression for the integral in Eq. (19) is only feasible due to the normal distributions involved. However, the computational cost of GPR is often hidden in this kernel optimization step since it requires solving linear systems and computing determinants [ref. 29, Sec. 2.3]. An appropriate criterion for stopping kernel optimizations that detects stagnant hyperparameters during an experiment is therefore provided below. Furthermore, observe that, for a fixed non-optimized kernel, $\sigma_J^2$ does not depend on observations $\hat{\mathbf{f}}_J$ but only on locations $X_J$ they have been made at.

The posterior mean function $m_J : \mathcal{X} \to \mathbf{R}$, incorporating knowledge on $J$ noisy observations of the function of interest $f$, can now be used as an approximation to $f$, whereas the posterior variance function

$\sigma_J^2 : \mathcal{X} \to [0,\infty)$ quantifies the corresponding uncertainties (Supplementary Fig. 5). Note that $\sigma_J^2(\mathbf{x}) < \sigma^2(\mathbf{x})$ for each $\mathbf{x} \in \mathcal{X}$ since $[\kappa_J(X_J, X_J) + \text{diag}(e(X_J))^2]^{-1}$ is positive-definite. Since we have $m \equiv 0$ later, we can further simplify Eq. (17) to

$$m_J(\mathbf{x}) = \kappa_J(\mathbf{x}, X_J)^\top \left[ \kappa_J(X_J, X_J) + \text{diag}(e(X_J))^2 \right]^{-1} \hat{\mathbf{f}}_J. \tag{20}$$

### Log-Gaussian processes

The Gaussian process, which is fitted to logarithmic intensity observations in our approach, is exponentiated to the original linear scale in this section to become log-Gaussian. Before describing details of its application in the TAS setting, we first give the relevant definitions and mention a technical detail that will be important below.

**Definition 3.** (Log-normal distribution[39]). Let $\eta \sim \mathcal{N}(0,1)$. Then, for parameters $\mu \in \mathbf{R}$ and $\sigma > 0$, the random variable

$$Z = \exp(\mu + \sigma\eta) \tag{21}$$

is said to follow a *log-normal distribution*, denoted by $Z \sim \log\text{-}\mathcal{N}(\mu, \sigma^2)$.

The mean and variance of a log-normally distributed random variable $Z$ are given by

$$\mathbf{E}[Z] = \exp\left(\mu + \frac{\sigma^2}{2}\right) \quad \text{and} \quad \mathbf{V}\text{ar}(Z) = (\exp(\sigma^2) - 1) \cdot \exp(2\mu + \sigma^2). \tag{22}$$

Below, we look at the noise distribution of log-Gaussian processes in order to satisfy our assumption on intensity observations to contain normally distributed noise (Eq. (1)). The following mathematical result is fundamental for this as it establishes a link between normal and log-normal distributions. It is proved in Supplementary Note 6.

**Proposition 1.** (Small variance limit for normalized log-normally distributed random variables) Let $Z$ be a log-normally distributed random variable as in Eq. (21) and define the corresponding normalized random variable

$$\overline{Z} := \frac{Z - \mathbf{E}[Z]}{\sqrt{\mathbf{V}\text{ar}(Z)}}. \tag{23}$$

Then, the random variable $\overline{Z}$ converges pointwise to $\eta$ as $\sigma \to 0^+$, i.e.,

$$\overline{Z}(\omega) \to \eta(\omega) \quad \text{as } \sigma \to 0^+ \tag{24}$$

for each $\omega \in \Omega$, where $\Omega$ denotes the sample space.

As mentioned, the exponentiation of a Gaussian process is log-Gaussian, by definition.

**Definition 4.** (Log-Gaussian process). Let $F$ be a Gaussian process. Then, the random process $(I(\mathbf{x}))_{\mathbf{x} \in \mathcal{X}}$ with

$$I(\mathbf{x}) = \exp(F(\mathbf{x})) \tag{25}$$

is called a *log-Gaussian process*.

Using Eqs. (17) and (18), it immediately follows for the posterior log-Gaussian process that

$$I(\mathbf{x}) \mid (\hat{F}(X_J) = \hat{\mathbf{f}}_J) \sim \log\text{-}\mathcal{N}(m_J(\mathbf{x}), \sigma_J^2(\mathbf{x})). \tag{26}$$

In particular, its posterior mean function is

$$\mathbf{E}[I(\mathbf{x}) \mid \hat{F}(X_J) = \hat{\mathbf{f}}_J] = \exp\left(m_J(\mathbf{x}) + \frac{\sigma_J^2(\mathbf{x})}{2}\right) \qquad (27)$$

and its posterior variance function becomes

$$\mathbf{Var}(I(\mathbf{x}) \mid \hat{F}(X_J) = \hat{\mathbf{f}}_J) = (\exp(\sigma_J^2(\mathbf{x})) - 1) \cdot \exp(2m_J(\mathbf{x}) + \sigma_J^2(\mathbf{x})). \qquad (28)$$

## Application to TAS

**Log-Gaussian processes for TAS.** In the context of our methodology from the previous sections, we choose the function of interest to be the logarithm of the intensity function from the TAS setting (Supplementary Note 1), i.e.,

$$f = \log i. \qquad (29)$$

If we define

$$\hat{F} = : \log \hat{I}, \qquad (30)$$

relating to Eq. (25), i.e., $F = \log I$, Eq. (13) gives

$$\begin{aligned} \log \hat{I}(\mathbf{x}_j) &= \log I(\mathbf{x}_j) + e(\mathbf{x}_j)\eta_j \\ \iff \hat{I}(\mathbf{x}_j) &= I(\mathbf{x}_j) \cdot \exp(e(\mathbf{x}_j)\eta_j) \end{aligned} \qquad (31)$$

for measurement locations $\mathbf{x}_j \in \mathcal{X}^*$. Note that, in contrast to the process $\hat{F}$ containing additive normally distributed noise (Eq. (13)), the noise of $\hat{I}$ is multiplicative and log-normal, i.e.,

$$\hat{I}(\mathbf{x}_j) \mid (I(\mathbf{x}_j) = i(\mathbf{x}_j)) = i(\mathbf{x}_j) \cdot \exp(e(\mathbf{x}_j)\eta_j). \qquad (32)$$

However, referring to Eq. (1), the physical assumption on the noise of intensity observations $I^+$ is to be additive and normally distributed, i.e.,

$$I^+(\mathbf{x}_j) \mid (I(\mathbf{x}_j) = i(\mathbf{x}_j)) = i(\mathbf{x}_j) + e^+(\mathbf{x}_j)\eta_j^+, \qquad (33)$$

where $\eta_j^+ \sim \mathcal{N}(0,1)$ and $e^+(\mathbf{x}_j) = \sqrt{i(\mathbf{x}_j)}$. Fortunately, the actual deviation of the two different noise distributions is negligible for sufficiently large intensities $i(\mathbf{x}_j)$ as the following explanations demonstrate.

As a first step, let us determine $e(\mathbf{x}_j)$ from Eq. (31) such that the variances of both noise distributions are equal, i.e.,

$$\begin{aligned} \mathbf{Var}(\hat{I}(\mathbf{x}_j) \mid I(\mathbf{x}_j) = i(\mathbf{x}_j)) &= \mathbf{Var}(I^+(\mathbf{x}_j) \mid I(\mathbf{x}_j) = i(\mathbf{x}_j)) \\ \iff i(\mathbf{x}_j)^2 \cdot (\exp(e(\mathbf{x}_j)^2) - 1) \cdot \exp(e(\mathbf{x}_j)^2) &= i(\mathbf{x}_j), \end{aligned} \qquad (34)$$

which yields

$$e(\mathbf{x}_j)^2 = \log\left(\frac{1}{2}\left(\sqrt{4/i(\mathbf{x}_j) + 1} + 1\right)\right). \qquad (35)$$

Note that the intensity term $i(\mathbf{x}_j)$ in Eq. (35) is not known in practice but can be replaced by the corresponding observation $\hat{i}_j \approx i(\mathbf{x}_j)$.

Since we aim to apply the small variance limit for log-normally distributed random variables from above (Eq. (24)), we set

$$Z = \hat{I}(\mathbf{x}_j) \mid (I(\mathbf{x}_j) = i(\mathbf{x}_j)) \sim \log{-}\mathcal{N}(\mu, \sigma^2) \qquad (36)$$

with

$$\mu = \log i(\mathbf{x}_j) \quad \text{and} \quad \sigma^2 = \log\left(\frac{1}{2}\left(\sqrt{4/i(\mathbf{x}_j) + 1} + 1\right)\right). \qquad (37)$$

Observe that

$$\sigma \to 0^+ \quad \text{as } i(\mathbf{x}_j) \to \infty. \qquad (38)$$

Furthermore, note that the term $\mu$ also depends on the limit $i(\mathbf{x}_j) \to \infty$ and hence on $\sigma \to 0^+$, in contrast to the setting above, but disappears in the calculations due to cancellations (see proof in Supplementary Note 6). Using the small variance limit, we immediately get the following result (Supplementary Fig. 6).

**Proposition 2.** The normalization of the noise random variable $\hat{I}(\mathbf{x}_j) \mid (I(\mathbf{x}_j) = i(\mathbf{x}_j))$ converges pointwise to $\eta$ as $i(\mathbf{x}_j) \to \infty$. In particular, it converges in distribution to a standard normal distribution.

If we set

$$\hat{\mathbf{i}}_J := (\hat{i}_1, \ldots, \hat{i}_J)^\top \in [0,\infty)^J, \qquad (39)$$

the acquisition function $\mathrm{acq}_J$ of our approach can now be defined as

$$\mathrm{acq}_J(\mathbf{x}) := \sqrt{\mathbf{Var}(I(\mathbf{x}) \mid \hat{I}(X_J) = \hat{\mathbf{i}}_J)}, \qquad (40)$$

which eventually gives Eq. (5) through Eq. (28).

**Intensity threshold and background level.** The intensity threshold $\tau$ and background level $\gamma$ have already been introduced informally in the Results section. Taking both into account as explained, the log-Gaussian process actually aims to approximate the modified intensity function

$$i_{\gamma,\tau}(\mathbf{x}) := \max\{\min\{i(\mathbf{x}),\tau\} - \gamma, 0\}. \qquad (41)$$

Observe that, by regarding intensity observations adjusted according to Eq. (41), the assumption of their noise being normally distributed is violated in general as their noise distribution is asymmetric. In particular, if $i(\mathbf{x}_j)$ substantially exceeds the intensity threshold $\tau$, the distribution gets rather concentrated at $\tau$ and thus small in variance. We, however, do assume noise on adjusted intensities as if they were observed so without adjusting. This does not change the expected behaviour of getting a useful acquisition function but even ensures numerical stability since noise regularizes the computational problem of solving linear systems in GPR.

It remains to explain how we seek to compute suitable values for the background level and intensity threshold without knowing the intensity function. We estimate $\gamma = \gamma(\mathcal{A}_0)$ and $\tau = \tau(\mathcal{A}_0)$ by statistics of $J_0 \in \mathbf{N}$ initial measurement points

$$\mathcal{A}_0 := ((\mathbf{x}_1, \hat{i}_1), \ldots, (\mathbf{x}_{J_0}, \hat{i}_{J_0})) \qquad (42)$$

collected to initialize our approach. The arrangement of initial measurement locations is specified below.

For computing the background level $\gamma$, we divide the initial intensity observations sorted in ascending order into 10 buckets

$$B_l := \{\hat{i}_j \mid D_{l-1} < \hat{i}_j \le D_l, j = 1, \ldots J_0\}, \quad l = 1, \ldots, 10, \qquad (43)$$

where $D_l$, $l = 1, \ldots, 9$, denotes the $l$-th decile of initial intensity observations, $D_0 := -\infty$, and $D_{10} := +\infty$. The relative and absolute differences of the bucket medians, i.e.,

$$\Delta_l^{\mathrm{rel}} := \frac{m_{l+1} - m_l}{m_l} \quad \text{and} \quad \Delta_l^{\mathrm{abs}} := m_{l+1} - m_l \qquad (44)$$

with $m_l := \mathrm{median}(B_l)$, $l = 1, \ldots, 9$, are taken to select the first bucket median which has a sufficiently large (relative and absolute) difference to its successor provided the corresponding decile does not exceed a

maximum decile. That is, we define

$$l^* := \min\left(\left\{l \in \{1,\ldots,9\} \mid \Delta_l^{\text{rel}} > \Delta_{\max}^{\text{rel}} \;\wedge\; \Delta_l^{\text{abs}} \geq \Delta_{\min}^{\text{abs}}\right\} \cup \{l_{\max}\}\right) \quad (45)$$

for parameters $\Delta_{\max}^{\text{rel}} > 0$, $\Delta_{\min}^{\text{abs}} > 0$, and $l_{\max} \in \{1,\ldots,9\}$ and set the function for computing the background level to

$$\gamma(\mathcal{A}_0) := m_{l^*}. \quad (46)$$

The intensity threshold $\tau$ is then selected as a value between the background level $\gamma$ and the maximum bucket median $m_{10}$ on a linear scale by the parameter $\beta$ that was introduced in the Results section. Therefore, we define

$$\tau(\mathcal{A}_0) := \gamma(\mathcal{A}_0) + \beta \cdot (m_{10} - \gamma(\mathcal{A}_0)). \quad (47)$$

Note that this definition is, by the meaning of $m_{10}$, robust to outliers in $\mathcal{A}_0$.

**Initial measurement locations.** The initial measurement locations $\mathbf{x}_1,\ldots,\mathbf{x}_{J_0} \in \mathcal{X}^*$ are deterministically arranged as a certain grid. It is chosen to be a variant of a regular grid in which every other row (or column) of points is shifted to the center of surrounding points (Supplementary Fig. 7). The intensity observations start in the bottom left corner of $\mathcal{X}^*$ and then continue row by row. Initial locations not reachable by the instrument, i.e., outside of $\mathcal{X}^*$, are skipped. If all initial locations are reachable, we use a total number of

$$J_0 := \frac{N_{\text{row}}^2 + 1}{2}, \quad (48)$$

where $N_{\text{row}} \in \mathbf{N}$ is the odd number of rows in the grid.

**Consumed areas.** As placing measurement points too close to each other has limited benefit for an efficient experiment, we mark areas around each measurement location $\mathbf{x}_j \in \mathcal{X}^*$ as "consumed" and ignore them as potential locations for new measurement points. Since the resolution function of a TAS yields ellipsoids in **Q**-$E$ space, it is natural to consider consumed areas in $\mathcal{X}^*$ as ellipses. An ellipse with center point $\mathbf{x}_j \in \mathcal{X}^*$ is defined as

$$\mathcal{E}(\mathbf{x}_j) := \{\mathbf{x} \in \mathbf{R}^n \mid (\mathbf{x} - \mathbf{x}_j)^\top E(\mathbf{x}_j)(\mathbf{x} - \mathbf{x}_j) \leq 1\} \quad (49)$$

for a matrix-valued function

$$E(\mathbf{x}_j) = U(\mathbf{x}_j)R(\mathbf{x}_j)^{-\top}R(\mathbf{x}_j)^{-1}U(\mathbf{x}_j)^\top = U(\mathbf{x}_j)R(\mathbf{x}_j)^{-2}U(\mathbf{x}_j)^\top \in \mathbf{R}^{n \times n}, \quad (50)$$

where $U(\mathbf{x}_j) \in \mathbf{R}^{n \times n}$ is a rotation matrix and $R(\mathbf{x}_j) = \text{diag}(r_1(\mathbf{x}_j),\ldots,r_n(\mathbf{x}_j)) \in \mathbf{R}^{n \times n}$ with $r_k > 0$. Then, the union of all ellipses at step $J$ is denoted by

$$\mathcal{E}_J := \bigcup_{j=1}^{J} \mathcal{E}(\mathbf{x}_j). \quad (51)$$

It is included in the objective function $\phi_J$ which is part of the final algorithm.

## Final algorithm

Incorporating all of the discussed methodological components, the steps for the final algorithm are listed in Box 2. Required components that have not been mentioned above are described in the next paragraphs. The algorithmic setting, i.e., particular values for parameters of the algorithm, is specified below.

**Objective function.** Recall that the objective function $\phi_J$, which is supposed to indicate the next measurement location, is composed of

the acquisition function $\text{acq}_J$ from Eq. (40) and the cost function $c_J$ from Eq. (3). In order to avoid distorting the objective function with physical units of time, we use the normalized cost function

$$\bar{c}_J(\mathbf{x}) := \frac{c_J(\mathbf{x})}{c_0}, \quad (52)$$

where $c_0 > 0$ is a normalizing cost value and set to the maximum distance between two initial measurement locations w.r.t. the metric $d$, i.e.,

$$c_0 := \max_{1 \leq j,j' \leq J_0} d(\mathbf{x}_j, \mathbf{x}_{j'}). \quad (53)$$

The objective function is then defined as

$$\phi_J(\mathbf{x}) := \begin{cases} \text{acq}_J(\mathbf{x})/(\bar{c}_J(\mathbf{x}) + 1) & \text{if } \mathbf{x} \notin \mathcal{E}_J, \\ -1 & \text{otherwise}. \end{cases} \quad (54)$$

Observe that $\phi_J$ excludes consumed areas in $\mathcal{E}_J$ as potential locations for new observations. Outside $\mathcal{E}_J$, it reflects the fact that the objective function should increase if the cost function decreases and vice versa. Also, if there were no costs present, i.e., $c_J \equiv 0$, then $\phi_J = \text{acq}_J$ outside $\mathcal{E}_J$.

**Kernel and optimizer for hyperparameters.** The parameterized kernel $\kappa_\theta$ is chosen to be the Gaussian (or radial basis function, RBF) kernel, i.e.,

$$\kappa_\theta(\mathbf{x},\mathbf{x}') := \sigma^2 \exp\left(-\frac{1}{2}(\mathbf{x} - \mathbf{x}')^\top \Lambda^{-1}(\mathbf{x} - \mathbf{x}')\right), \quad (55)$$

where $\sigma^2 > 0$ and $\Lambda = \text{diag}(\lambda_1,\ldots,\lambda_n) \in \mathbf{R}^{n \times n}$ for length scales $\lambda_k \geq 0$. Hence, the vector of hyperparameters is

$$\theta = (\sigma^2, \lambda_1, \ldots, \lambda_n)^\top \in \mathbf{R}^{n+1}. \quad (56)$$

Recall that the kernel is needed for GPR to fit a Gaussian process to the logarithm of intensity observations. As mentioned above, we compute optimal hyperparameters by maximizing the logarithm of the marginal likelihood (Eq. (19)). Since this optimization problem is non-convex in general, it might have several local maxima. Instead of a global optimizer, we run local optimizations starting with $N_\mathcal{H} \in \mathbf{N}$ different initial hyperparameter values distributed uniformly at random in a hypercube $\mathcal{H} \subseteq \mathbf{R}^{n+1}$ and choose the one with the largest local maxima. Note that this introduces stochasticity into our approach.

**Stopping criterion for kernel optimizations.** As kernel optimizations are computationally the most expensive part of our methodology, it is reasonable to stop them once a certain criterion is met. The stopping criterion $P_{\text{KO}} = P_{\text{KO}}(J)$ is formalized as a predicate, i.e., a boolean-valued function, depending on step $J$. If $N_{\text{KO}}(J) \in \mathbf{N}$ denotes the number of kernel optimizations performed up until step $J$, it is defined as

$$P_{\text{KO}}(J) := N_{\text{KO}}(J) > N_{\text{KO}}^{\max}$$
$$\vee \left(N_{\text{KO}}(J) \geq N_{\text{KO}}^{\min} \;\wedge\; \frac{1}{k_{\text{KO}} - 1}\sum_{j=J-k_{\text{KO}}+1}^{J-1}\frac{\|\log(\theta_j) - \log(\theta_{j+1})\|_2}{\|\log(\theta_{j+1})\|_2} \leq \varepsilon_{\text{KO}}\right) \quad (57)$$

for parameters $N_{\text{KO}}^{\min}, N_{\text{KO}}^{\max}, k_{\text{KO}} \in \mathbf{N}$ such that

$$2 \leq k_{\text{KO}} \leq N_{\text{KO}}^{\min} \leq N_{\text{KO}}^{\max} \quad (58)$$

and $\varepsilon_{\text{KO}} > 0$. Informally, this predicate indicates that kernel optimizations should be stopped as soon as $N_{\text{KO}}(J)$ exceeds a given maximum

number $N_{KO}^{max}$ or if, provided that $N_{KO}(j)$ exceeds a given minimum number $N_{KO}^{min}$, the average relative difference of the last $k_{KO}$ hyperparameters falls below a given threshold value $\varepsilon_{KO}$, i.e., the hyperparameters stagnate and do no longer change substantially. Note that, in Eq. (57), the expressions $\log(\theta)$ for the vector of kernel hyperparameters $\theta = (\sigma^2, \lambda_1, \dots, \lambda_n)^\top$ from Eq. (56) are meant componentwise, i.e.,

$$\log(\theta) = (\log(\sigma^2), \log(\lambda_1), \dots, \log(\lambda_n))^\top \in \mathbf{R}^{n+1}. \quad (59)$$

**Cost measure.** Finally, the cost measure $c$ is chosen to represent experimental time, i.e., the total time needed to carry out an experiment $\mathcal{A}$. Experimental time consists of the cumulative counting time and the cumulative time for moving the instrument axes. The cumulative counting time is measured by

$$c_{count}(\mathcal{A}) := \sum_{j=1}^{|\mathcal{A}|} T_{count,j} \quad (60)$$

where $T_{count,j} \geq 0$ denotes the single counting time, i.e., the time spent for a single intensity observation, at $\mathbf{x}_j$. The cost measure for the cumulative time spent to move the instrument axes is defined as

$$c_{axes}(\mathcal{A}) := \sum_{j=1}^{|\mathcal{A}|-1} d(\mathbf{x}_j, \mathbf{x}_{j+1}), \quad (61)$$

where $d$ is a metric representing the cost for moving from $\mathbf{x}_j$ to $\mathbf{x}_{j+1}$. For details, we refer to Supplementary Note 1 (Eq. (11)). Eventually, we set

$$c(\mathcal{A}) := c_{count}(\mathcal{A}) + c_{axes}(\mathcal{A}). \quad (62)$$

For simplicity, the single counting times are assumed to be constant on the entire domain $\mathcal{X}^*$, i.e., $T_{count,j} = T_{count} \geq 0$ yielding

$$c_{count}(\mathcal{A}) = |\mathcal{A}| \cdot T_{count}. \quad (63)$$

## Degenerate cases

An intensity function that is nearly constant along a certain coordinate $x_k$ in $\mathcal{X}$, i.e., an intrinsically lower-dimensional function, might cause problems for the Gaussian kernel from Eq. (55) as the corresponding optimal length scale hyperparameter would be $\lambda_k = \infty$. Also, the initial observations $\mathcal{A}_0$ from Eq. (42) might not resolve the main characteristics of the intensity function sufficiently well and hence pretend it to be lower-dimensional.

Most degenerate cases can be identified by kernel optimizations resulting in one or more length scales that are quite low or high relative to the dimensions of $\mathcal{X}$. In general, we assess a length scale parameter $\lambda_k$ as degenerate if it violates

$$\delta^- \leq \frac{\lambda_k}{x_k^+ - x_k^-} \leq \delta^+ \quad (64)$$

for two parameters $0 < \delta^- < \delta^+ < \infty$, where $x_k^-$ and $x_k^+$ denote the limits of the rectangle $\mathcal{X}$ in dimension $k$. If, after kernel optimization at a certain step, a length scale parameter is recognized to be degenerate, we regard the intensity function on a coordinate system rotated by 45° in order to avoid the mentioned problems. Of course, the rotation is performed only internally and does not affect the original setting.

In $n = 2$ dimensions, a crystal field excitation, for example, might induce a lower-dimensional intensity function (Supplementary Fig. 8a). After rotating the coordinate system, the intensity function becomes full-dimensional (Supplementary Fig. 8b) allowing non-degenerate kernel optimizations.

## Algorithmic setting

All experiments described in this article are performed in $n = 2$ dimensions, i.e., $\mathcal{X} \subseteq \mathbf{R}^2$. If not specified otherwise, we use $N_{row} = 11$ rows corresponding to 61 measurements in the initialization grid (Eq. (48)). For scenario 2 of the neutron experiment, we use $N_{row} = 7$ rows corresponding to 25 initial measurements. The default parameter values used for the final algorithm (Box 2) in both experimental settings, i.e., the neutron experiment and the benchmark, are specified in Table 1.

Although NICOS is able to consider an instrument's resolution function for the computation of matrices $E(\mathbf{x}_j)$ defining ellipses as consumed areas, we decided to use ellipses fixed over $\mathcal{X}^*$ for both, the neutron experiment and the benchmark. Thus, the function $E$ (Eq. (50)) is chosen to give circles with fixed radius $r > 0$ on a normalized domain, i.e., $U(\mathbf{x}_j) = I$ and

$$r_k(\mathbf{x}_j) = \frac{r}{x_k^+ - x_k^-}. \quad (65)$$

We set $r = 0.02$ for the neutron experiment and $r = 0.025$ for the benchmark.

## Data availability

Source data for figures and movies related to the neutron experiment or the benchmark are available in the repository at jugit.fz-juelich.de/ainx/ariane-paper-data.

## Code availability

The software implementation of our approach is based on the GaussianProcessRegressor class from the Python package scikit-learn[40]. All results can be reproduced using our code from the repository jugit.fz-juelich.de/ainx/ariane (commit SHA: c1c31c96). The benchmark results can be reproduced by using code from the repository jugit.fz-juelich.de/ainx/base-fork-ariane (commit SHA: 3715a772). It is a fork, adjusted to our approach, of the benchmark API from jugit.fz-juelich.de/ainx/base which is part of the mentioned original work on the benchmarking procedure[34].

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

## Acknowledgements

This work was supported through the project *Artificial Intelligence for Neutron and X-ray scattering* (AINX) funded by the Helmholtz AI cooperation unit of the German Helmholtz Association. We thank Yuliia Tymoshenko (KIT) for providing simulations on $ZnCr_2Se_4$ and FeP as well as Anup Bera and Bella Lake (both HZB) for the same on $SrCo_2V_2O_8$. Furthermore, we gratefully thank Frank Weber (KIT) who provided the SnTe sample and related information for the neutron experiment at EIGER. We would also like to thank Kuangdai Leng (STFC) for his suggestion to include a schematic representation of our approach (Fig. 2). Eventually, we acknowledge the discussions and collaborations with Marcus Noack (LBNL) and Martin Boehm (ILL).

## Author contributions

M.G. and A.S. initiated and supervised the research. M.T.P. developed the methodology. M.T.P. and G.B. designed and developed the software components. M.T.P., G.B., C.F., U.S., and A.S. performed the experiments. M.T.P. wrote the first draft of the manuscript. All authors revised drafts of the manuscript.

## Funding

## Competing interests

The authors declare no competing interests.
