## [Peer Review File · Nature Communications]

Active learning-assisted neutron spectroscopy with
log-Gaussian processesREVIEWER COMMENTS

Reviewer #1 (Remarks to the Author):

The authors report on the use of log-Gaussian processes for automated/on-the-fly experiment planning.

Active learning has the potential of optimizing measurements to minimize acquisition time and maximize information content, which is particularly important for neutron scattering techniques where beam availability is limited. The authors demonstrate their approach with measurements on a neutron triple-axis instrument.

This field has seen great attention in recent years. This manuscript is clearly articulated and presents new and useful results. I found the manuscript to be very good, and I only have minor comments. It should be published with minor revisions.

- Line 85: The authors mention that mixing the standard deviation and the mean of a Gaussian process creates risks of losing some of its interpretability. Could the authors expand on this statement? It's not quite clear but could be an important consideration.

- Fig 3: I would recommend expanding the $(h,h,3)$ axis to the left and right so we can clearly see the points at $h=0$ and $h=2$. It was initially difficult for me to understand why the sharp feature at $h=2$ was visible in Row 1 before realizing that there were points on the edge of the figure.

Reviewer #2 (Remarks to the Author):

Please find the attached PDF.

Review comments to
*AI-assisted neutron spectroscopy using active
learning with log-Gaussian processes*

Kuangdai Leng

November 2022

This manuscript presents an application of the Gaussian Process Regression (GPR) to neutron experiments at three-axes spectrometers (TAS) in an attempt to maximise beamtime utilisation. The main contribution is an automated, on-the-fly sampling scheme that determines the next measurement point based on the existing measurements by maximising a gain over cost, taking into account several experimental characteristics of TAS data. The authors have shown that their scheme is more efficient than conventional grid sampling and Monte Carlo sampling.

In general, I think the research work presented here presumably qualifies publication in Nat Comms given its potentiality to uplift TAS experiments for new scientific discoveries. However, the manuscript, in its current form, reads inadequately expressing and convincing to me from a technical viewpoint. My recommendation is **Major Revisions**. In the reminder of this report, I will focus on three issues, ordered by significance.

1 Method comparison

The most outstanding insufficiency to me is that the proposed GPR-based scheme is only compared with two very basic schemes: structured grid and Monte Carlo. Completely ignoring acquired knowledge, these two schemes are literally the “baselines”, which cannot justify the superiority of the purposed method. I give my two reasons below.

First, known from the introduction, the authors are aware of quite a few autonomous approaches applied to X-ray or neutron scattering; among them, GPcam has been implemented for TAS. Therefore, it is natural for a reader to expect a comparison between the proposed one and the state-of-the-art (SOTA), which, to me, is GPcam. Even if the authors are unable to run GPcam on their data, for any possible reason, the authors can still compare the following variants of their scheme (so no need to use any other code):

- intensity in linear scale (GPcam) vs. log scale (this work);

- mean-variance-based (GPcam) vs. variance-based (this work, eq.(43)) acquisition functions.

These are the two major differences between GPcam and the proposed method. The authors claim such differences as technical novelties but showing no data-based evidence to support their advantage. In the current version, the authors are focused on the effects of intensity threshold τ and background level γ , which are important to their method, of course, but not comparable to previous works in the course of showing SOTA.

Second, between the GPcam-like studies and the two baseline methods, there still exists a huge gap. For instance, rather than using GPR or any other Bayesian techniques, one can choose x_{J+1} in Step 4, Algorithm 1 based on some empirical principles, such as performing denser sampling within regions where the gradients of the current intensity function are larger – a common practice known as gradient-guided adaptive sampling. An empirical scheme as such can have many side effects, but it is still very likely to be more efficient than structured grid and Monte Carlo.

In summary, I think there are four levels of comparable techniques:

1. SOTA in TAS or similar methods implemented for neutron scattering, such as GPcam;
2. some variants of the proposed method more or less equivalent to some previous methods (e.g., GPR with linear intensity);
3. some empirical or non-Bayesian schemes within the framework of Algorithm 1;
4. structured grid and Monte Carlo.

Certainly, I am NOT suggesting that the authors should do all above, which can probably justify another paper. However, the higher level they can archive, the more compelling this paper becomes. Here my key point is that Level 4 alone can hardly prove SOTA or support any advancement this work has created w.r.t. previous studies.

2 Method delivery

From my reading, the whole method is not delivered in a very efficient manner. In the current manuscript, the authors try to describe the most critical definitions and techniques in the first three subsections of *Results*, detailing them in *Methods* and leaving insignificant details to *Supplementary*. This arrangement is reasonable, but the delivery is unclear. For *Results*, the whole framework seems not well established in *Problem formulation*, and how the following two subsections (which contain quite a bit details) are connected to *Problem formulation* is not explicitly explained. This fragmented method description is then followed by the experimental results and discussions. *Methods* starts with the

Figure 1: Method framework to my understanding. This can be inaccurate.

basics of GPR and Log-GP, presenting a forest of equations that seems quite distracting to me. The key information comes before and after Algorithm 2 (such as eqs.(43) and (60)), and the whole picture of what the authors have done are finally built in my mind only upon the last second page (given that I am fairly familiar with Bayesian, GP and kernel machines).

To make myself clear, the “whole picture” I was mentioning earlier is sketched in Figure 1 in a top-down manner. In my opinion, it is better to drive home this picture as early as possible before going into any details, including what each component is for and how they are connected. Also, any equations or technical details that are non-essential for the understanding of this picture can be moved to *Supplementary* for better readability, especially those not originating from this work. Currently, the authors describe some segments of this picture in *Results* (such as τ and γ), as they are required for reporting the experimental results, leaving all the rest in *Methods* in a flat organisation and probably with an unbalanced verbosity.

3 A more comprehensive introduction

This point is less significant than the previous two. The current introduction seems not providing a comprehensive and understandable overview of the back-

ground, challenges and novelties of this paper for a wider audience. For people with a background in data science or scientific computing (such as me), the basics and significance of neutron TAS are not explained (in particular, why it requires many measurements in an high-dimensional space and why such a space has a varying information density). For neutron scientists (especially beam scientists), GP-related techniques such as Bayesian inference with MCMC or HMC are not introduced. Of course, the authors do write one or two sentences around those points, if that counts. The current style of introduction is more frequently seen in a domain-specific journal, running fast into tech specs at the beginning and discussing one or two specific issues in great detail (e.g., GPcam). However, as the authors choose to submit to Nat Comms, they should better target a wider audience with either AI or neutron background, or even neither.

Response to reviews of

AI-assisted neutron spectroscopy using active learning with log-Gaussian processes

M. Teixeira Parente, G. Brandl, C. Franz, U. Stuhr, M. Ganeva, and A. Schneidewind

January 20, 2023

We would like to thank the two reviewers who took the time to carefully read our manuscript and provide useful suggestions and feedback that we believe have improved the manuscript. Our responses to their specific points are given below between the lines first. The main changes related to the reviewers' comments as well as general changes (not related to the reviewers' comments) that we have made for the revised manuscript are summarized at the end.

Review #1

The authors report on the use of log-Gaussian processes for automated/on-the-fly experiment planning. Active learning has the potential of optimizing measurements to minimize acquisition time and maximize information content, which is particularly important for neutron scattering techniques where beam availability is limited. The authors demonstrate their approach with measurements on a neutron triple-axis instrument.

This field has seen great attention in recent years. This manuscript is clearly articulated and presents new and useful results. I found the manuscript to be very good, and I only have minor comments. It should be published with minor revisions.

Dear Reviewer #1,

We thank you very much for your review and really appreciate your comments and feedback.

- *Line 85: The authors mention that mixing the standard deviation and the mean of a Gaussian process creates risks of losing some of its interpretability. Could the authors expand on this statement? It's not quite clear but could be an important consideration.*

We sorted out what we mean by claiming that “gpCAM risks losing some its interpretability” and added clarifying elements in the sentence thereafter (lines 96 – 99).

Lines 96 – 99 (references cited in the original text were removed here): “*Additionally, the acquisition function used in the only TAS experiment of gpCAM so far makes it risk losing some of its interpretability in the TAS setting, because physical units do not match and it is not obvious from a physics point of view why this function in particular is suitable for the goal of placing measurement points in regions of signal. In contrast, we are able to choose a particular canonical acquisition function that remains the same for each experiment since we identify regions of signal by the methodology itself or, more concretely, by the logarithmic transformation of a Gaussian process, and not primarily by an acquisition function as the critical component.*”

- *Fig 3: I would recommend expanding the $(h, h, 3)$ axis to the left and right so we can clearly see the points at $h=0$ and $h=2$. It was initially difficult for me to understand why the sharp feature at $h=2$ was visible in Row 1 before realizing that there were points on the edge of the figure.*

We feel that this is a very valuable suggestion and therefore changed Fig. 3 accordingly (Fig. 1 here). We additionally put a small black edge around the markers to improve their visibility. Furthermore, we adjusted all other figures that also display intensity functions with measurement points on the edges. Please note that we adjusted all supplementary videos as well.

Figure 1: New Fig. 3.

We hope that our responses are comprehensible and the changes help to address your concerns. We are looking forward to hear back from you.

Yours sincerely,

Mario Teixeira Parente, on behalf of the whole team of authors

Review #2

This manuscript presents an application of the Gaussian Process Regression (GPR) to neutron experiments at three-axes spectrometers (TAS) in an attempt to maximise beamtime utilisation. The main contribution is an automated, on-the-fly sampling scheme that determines the next measurement point based on the existing measurements by maximising a gain over cost, taking into account several experimental characteristics of TAS data. The authors have shown that their scheme is more efficient than conventional grid sampling and Monte Carlo sampling.

*In general, I think the research work presented here presumably qualifies publication in Nat Comms given its potentiality to uplift TAS experiments for new scientific discoveries. However, the manuscript, in its current form, reads inadequately expressing and convincing to me from a technical viewpoint. My recommendation is **Major Revisions**. In the remainder of this report, I will focus on three issues, ordered by significance.*

Dear Dr Kuangdai Leng,

We thank you very much for your review and really appreciate your comments and feedback since we believe that addressing your concerns has further improved the manuscript.

Certainly your main point is the presentation of a comparison with gpCAM and other approaches. We are confident to have found an appropriate solution for this request. More details, also to any other points that you have raised, can be found below.

1 Method comparison

The most outstanding insufficiency to me is that the proposed GPR-based scheme is only compared with two very basic schemes: structured grid and Monte Carlo. Completely ignoring acquired knowledge, these two schemes are literally the “baselines”, which cannot justify the superiority of the purposed method. I give my two reasons below.

We agree with you that the random and grid approach are both just “baselines”. However, in our opinion, any new approach should first clearly show that it performs significantly better than these baselines. Our responses regarding a comparison with gpCAM can be found below.

First, known from the introduction, the authors are aware of quite a few autonomous approaches applied to X-ray or neutron scattering; among them, GPcam has been implemented for TAS. Therefore, it is natural for a reader to expect a comparison between the proposed one and the state-of-the-art (SOTA), which, to me, is GPcam.

We also feel that it is natural to expect a comparison with gpCAM. However, before our first submission, we were not sure if this was feasible. gpCAM does not provide a full specification for values of its parameters in the TAS setting. For example, we did not know how to choose the acquisition function, its main parameter, and therefore initially decided not to run a comparison. Without a particular acquisition function for the TAS setting, that gpCAM commits to, i.e., using arbitrary acquisition functions, a comparison would produce meaningless results since the acquisition function is a very sensitive parameter of gpCAM. However, we have now decided to provide a comparison based on all the information available to us about gpCAM. That is, we have chosen an acquisition function that was formerly used for gpCAM for a neutron experiment at the instrument ThALES (ILL) [1, Eq. (6)]. In fact, we have contacted Marcus Noack (LBNL, responsible author of gpCAM and its related publications) via e-mail to make sure that our way of using gpCAM is correct and we do not make any mistakes. As he confirmed our usage of gpCAM, we assume that we have run gpCAM in the correct way and have produced reliable results. More details on the comparison can be found below, once the context of your review fits better.

However, we do not agree with your opinion that gpCAM is the state-of-the-art for the TAS setting. For claiming state-of-the-art for the TAS setting, an approach must provide reproducible evidence that it reliably performs reasonable experiments for a large variety of test cases, whether in a synthetic setting or in real-world neutron experiments. None of that is provided by gpCAM. The only reference that provides some sort of evidence is the aforementioned, so far only, neutron experiment at ThALES [1]. The intensity distribution in this case, however, has quite large regions of signal and is thus simple to detect for an approach such as gpCAM. Our comparison shows that

gpCAM works reasonably well in scenarios like this, but fails for more advanced test cases (at least with the parameters for gpCAM that we have used).

Even if the authors are unable to run GPcam on their data, for any possible reason, the authors can still compare the following variants of their scheme (so no need to use any other code):

- *intensity in linear scale (GPcam) vs. log scale (this work);*
- *mean-variance-based (GPcam) vs. variance-based (this work, eq.(43)) acquisition functions.*

These are the two major differences between GPcam and the proposed method. The authors claim such differences as technical novelties but showing no data-based evidence to support their advantage. In the current version, the authors are focused on the effects of intensity threshold τ and background level γ , which are important to their method, of course, but not comparable to previous works in the course of showing SOTA.

We are actually able to run gpCAM on our data and therefore let gpCAM run in our benchmark setting with three different variants for a comparison with our approach. However, we did not put the results in the Results section but in the Supplementary Information (Sec. S5). The reason for this is provided at the end of the Benchmark section (lines 305 – 311), but was basically already formulated above. For a well-formulated and sustainable comparison, there is just not enough information on how to run gpCAM in the TAS setting, i. e., on which acquisition function(s) should be considered.

We hope that our comparison meets your request. A more comprehensive and sustainable comparison with gpCAM is, in our opinion, not possible with the information currently available about gpCAM and the way it should be used.

You were also suggesting to compare our current approach with some variants (linear vs. log scale, mean-variance-based vs. variance-based acquisition functions). Our decision to approximate intensities on a logarithmic scale and then take the variance function (actually, the standard deviation function) of the corresponding log-Gaussian process as acquisition function is motivated by the scientific case of TAS experiments, i. e., the detection of regions of signal. This decision is fixed from the very beginning of our methodology and not a choice as for gpCAM. Therefore, there is no meaning behind varying the scale of collected intensity data and/or the acquisition function in the case of our approach. In our opinion, the fact that our acquisition function is not a choice but stays fixed for each experiment is a huge advantage over gpCAM’s free choice. The acquisition function for gpCAM is a part of its methodology but its choice should be motivated from an application point of view (here TAS). A missing connection between the acquisition function as a methodological part and the scientific objective of a field of application, as we believe is the case with gpCAM and TAS, therefore makes it very difficult to make a good choice for an acquisition function. For all of these reasons, we do not believe a comparison with your proposed variants is applicable here.

Second, between the GPcam-like studies and the two baseline methods, there still exists a huge gap. For instance, rather than using GPR or any other Bayesian techniques, one can choose x_{J+1} in Step 4, Algorithm 1 based on some empirical principles, such as performing denser sampling within regions where the gradients of the current intensity function are larger – a common practice known as gradient-guided adaptive sampling. An empirical scheme as such can have many side effects, but it is still very likely to be more efficient than structured grid and Monte Carlo.

We agree that there is a gap between approaches completely ignoring acquired knowledge and approaches based on Bayesian methodology. An empirical approach that would fill this gap and, as you described, places measurement points at locations with large gradients is however difficult to implement in the TAS setting, because it obviously relies on a sufficiently accurate approximation of the intensity function’s gradient. Such a reliable approximation requires quite some measurement points which are not available in a TAS experiment, especially not in the beginning where it is important to place measurement points in a reasonable manner. In fact, gpCAM was indeed using gradient information as part of its acquisition function in its early days (see, e. g., [2, Eq. (10)]). However, a private conversation with Marcus Noack revealed that the team refrained from this idea since the output did not meet their expectations in the long run. The reason, we think, lies in the poor approximation of the gradient with only a couple of data points.

In addition, an empirical approach as suggested does not account for the fact that measurement points should be reasonably distributed over regions of signal. It would place measurement points

in regions of large gradient, no matter how many measurement points are lying there already. An approach that accounts for this needs to quantify some sort of uncertainty which directly brings in a Bayesian thinking. Thus, in the TAS setting, the mentioned gap may not be as large as it seems at first glance. For these reasons, we refrain from making a corresponding comparison.

In summary, I think there are four levels of comparable techniques:

1. *SOTA in TAS or similar methods implemented for neutron scattering, such as GPcam;*
2. *some variants of the proposed method more or less equivalent to some previous methods (e.g., GPR with linear intensity);*
3. *some empirical or non-Bayesian schemes within the framework of Algorithm 1;*
4. *structured grid and Monte Carlo.*

Certainly, I am NOT suggesting that the authors should do all above, which can probably justify another paper. However, the higher level they can archive, the more compelling this paper becomes. Here my key point is that Level 4 alone can hardly prove SOTA or support any advancement this work has created w.r.t. previous studies.

Eventually, we agree with you that there needs to be more comprehensive comparisons of approaches for AI-assisted TAS experiments. This was exactly the reason why we have developed a corresponding benchmark which makes it possible to compare these approaches in the TAS setting [3].

But for reasons mentioned above, such a comprehensive comparison is currently unfortunately not possible. Nevertheless, we hope that you will appreciate the comparison of our approach with gpCAM in the Supplementary Information and that it will be an asset to the community.

2 Method delivery

From my reading, the whole method is not delivered in a very efficient manner. In the current manuscript, the authors try to describe the most critical definitions and techniques in the first three subsections of “Results”, detailing them in “Methods” and leaving insignificant details to “Supplementary”. This arrangement is reasonable, but the delivery is unclear. For “Results”, the whole framework seems not well established in “Problem formulation”, and how the following two subsections (which contain quite a bit details) are connected to “Problem formulation” is not explicitly explained. This fragmented method description is then followed by the experimental results and discussions. “Methods” starts with the basics of GPR and Log-GP, presenting a forest of equations that seems quite distracting to me. The key information comes before and after Algorithm 2 (such as eqs.(43) and (60)), and the whole picture of what the authors have done are finally built in my mind only upon the last second page (given that I am fairly familiar with Bayesian, GP and kernel machines).

To make myself clear, the “whole picture” I was mentioning earlier is sketched in Figure 2 in a top-down manner. In my opinion, it is better to drive home this picture as early as possible before going into any details, including what each component is for and how they are connected. Also, any equations or technical details that are non-essential for the understanding of this picture can be moved to “Supplementary” for better readability, especially those not originating from this work. Currently, the authors describe some segments of this picture in “Results” (such as τ and γ), as they are required for reporting the experimental results, leaving all the rest in “Methods” in a flat organisation and probably with an unbalanced verbosity.

We thank you very much for this comment since it made us think more carefully about how we want to present our approach. You raise three different points here:

1. The introduction of our methodology in “Results” and the missing connection to “Problem formulation”
2. The fragmented description of our methodology
3. The flat organization and the level of detail in “Methods”

We would like to answer to each point separately.

Figure 2: *Method framework to my understanding. This can be inaccurate.*

1. We agree with you that this point can be improved. Therefore, we adopted your suggestion of placing a schematic figure that shows the components of our approach and how they are connected as early as possible (now Fig. 2 in our manuscript, Fig. 3 here). We added related text at the end of “Problem formulation” (lines 165 – 170) for readability and feel that the connection of our methodology in the context of the problem formulation and Alg. 1 is now clearer. Of course, your suggestion to include a schematic representation of our approach is acknowledged under “Acknowledgements”.

Lines 165 – 170: “*Our methodology concentrates on developing a useful acquisition function as the crucial component of our approach making it straightforward to find regions of signal. A schematic representation with general and specific components of our approach in the context of Alg. 1 can be found in Fig. 2. Specific components necessary to understand the experimental results are informally introduced below. Details for these specific components as well as our final algorithm (Alg. 2), concretizing Alg. 1, along with all its parameters and their particular values used are specified in the Methods section.*”

2. In our opinion, the impression of a fragmented description of the method was caused by the subsection “Intensity threshold and background level” since we mentioned quite some details already there, which are not necessary for understanding the experimental results. For this reason, we shortened this subsection, leaving only necessary elements, and details were moved to the “Methods” section. The introduction of necessary components is now clearer and the reader is not distracted with details that are not needed to follow the text.

3. We agree with you that the flat organization of “Methods” makes the text less readable and harder to follow. Therefore, we added another section level for readability such that the new organization of subsections in “Methods” now looks like the following:

- Gaussian Process Regression
- Log-Gaussian processes
- Application to TAS
 - Log-Gaussian processes for TAS
 - Intensity threshold and background level
 - Initial measurement locations

Figure 3: Schematic representation of general (solid line boxes) and specific (dashed line boxes) components of our approach in the context of Alg. 1. The specific components are discussed in detail in the Methods section.

- Consumed areas
- Final algorithm
 - Objective function
 - Kernel and optimizer for hyperparameters
 - Stopping criterion for kernel optimizations
 - Cost measure
- Degenerate cases
- Algorithmic setting

Please note that we decided to move the paragraph on the estimation of the intensity threshold and background level as well as the paragraph on initial measurement locations (both formerly under “Final algorithm”) to sections “Intensity threshold and background level” and “Initial measurement locations”, respectively (now both under “Application to TAS”). Also, the section “Final algorithm” has got its own subsections.

Regarding the level of detail in “Methods”, we do understand your concern, but do not fully agree with you here. In fact, the “Brief guide for submission to *Nature Communications*” (available at [nature.com/documents/ncomms-submission-guide.pdf](https://www.nature.com/documents/ncomms-submission-guide.pdf)) clearly says on page 1: “*The Methods section [...] should contain all elements necessary for interpretation and replication of the results. Methods should be written as concisely as possible and ...*”. This is the reason why we decided to provide a decent amount of detail in “Methods”. In particular, the basics of GPR and log-GPs are relevant in order to understand why our approach works and produces the results we have shown, especially for a wider audience typical for *Nature Communications* that has no or not much knowledge about probabilistic function approximation and its statistical interpretation in a field of application such as TAS. However, we agree with you that there is quite a number of equations at the beginning of “Methods”, but feel that they are necessary for an appropriate notation and concise reasoning about our method as a whole. To reduce the distracting aspect of this set of equations, we had originally provided a summarizing table (Tab. 1) with all parameters of our approach and their values used under “Algorithmic setting” such that the reader can easily access this important information in a distilled and accessible form.

3 A more comprehensive introduction

This point is less significant than the previous two. The current introduction seems not providing a comprehensive and understandable overview of the background, challenges and novelties of this paper for a wider audience. For people with a background in data science or scientific computing (such as me), the basics and significance of neutron TAS are not explained (in particular, why it requires many measurements in an high-dimensional space and why such a space has a varying information density). For neutron scientists (especially beam scientists), GP-related techniques such as Bayesian inference with MCMC or HMC are not introduced. Of course, the authors do write one or two sentences around those points, if that counts. The current style of introduction is more frequently seen in a domain-specific journal, running fast into tech specs at the beginning and discussing one or two specific issues in great detail (e.g., GPcam). However, as the authors choose to submit to Nat Comms, they should better target a wider audience with either AI or neutron background, or even neither.

Thank you very much for this comment. We agree with you that *Nature Communications* has a broader audience than domain-specific journals and that we can better accommodate that. In particular, you suggest to

1. mention the basics and significance of neutron TAS for readers not familiar with TAS,
2. introduce GP-related techniques such as Bayesian inference with MCMC and HMC for neutron scientists.

We would like to address the two points separately.

1. We agree with you here. So, we restructured and extended the first paragraph of the introduction (now two paragraphs) to make it more accessible for readers not familiar with TAS (lines 25 – 38). Now, it should be clearer what the basics of TAS are, that TAS is a significant technique in fundamental materials research, and what causes the information density to be varying in \mathbf{Q} - E space.

Lines 25 – 38 (references cited in the original text were removed here): “*Neutron three-axes spectrometers (TAS) enable understanding the origins of materials properties through detailed studies of magnetic and lattice excitations in a sample. Developed in the middle of the last century, the technique remains one of the most significant in fundamental materials research and was therefore awarded the Nobel prize in 1994. It is used for investigating the most interesting and exciting phenomena of their time: the cause of different crystal structures in iron, unconventional superconductors, quantum spin glasses, quantum spin liquids, and non-trivial magnetic structures.*

TAS are globally operated at neutron sources of large-scale research facilities and measure scattering intensity distributions in the material’s four-dimensional \mathbf{Q} - E space, i. e., in its momentum space (\mathbf{Q}) for different energy transfers (E), by counting scattered neutrons on a single detector. However, high demand and limited availability make beam time at TAS a valuable resource for experimenters. Since, furthermore, the intensity distributions of the aforementioned excitations have an information density that strongly varies over \mathbf{Q} - E space due to the direction of the underlying interactions and the symmetry of the crystal, and TAS measure sequentially at single locations in \mathbf{Q} - E space, it is natural to think about if and how we can improve the efficiency of TAS experiments and make better use of the experimenter’s time.”

2. For readers not familiar with Gaussian processes, we added a sentence on GPR as a Bayesian technique in the fifth paragraph (lines 59 – 62). It should make clear that we use GPR in order to quantify a sort of uncertainty which, in our case, can be exploited to find regions of signal. We however do not understand why you suggest to introduce “techniques such as Bayesian inference with MCMC or HMC”, because we do not use Markov chain Monte Carlo methods in our approach at all. Also, Bayesian thinking, i. e., the updating of prior knowledge by an incorporation of empirical data, should be sufficiently well known in any empirical field of science nowadays. We therefore would like to refrain from introducing Bayesian inference and the basic concepts associated with it.

Lines 59 – 62: “*The fact that GPR is a Bayesian technique that uses pointwise normal distributions to fit noisy data from an underlying inaccessible function of interest makes it indeed an interesting candidate for many applications and use cases where it is important to quantify some sort of uncertainty.*”

We hope that our responses are comprehensible and the changes help to address your concerns. We are looking forward to hear back from you.

Yours sincerely,

Mario Teixeira Parente, on behalf of the whole team of authors

Main changes

Figures

- All figures displaying intensity functions with measurement points on the edges are adjusted such that measurement points on the edges are now visible.

Introduction

- First two paragraphs: We divided the former first paragraph into two paragraphs and extended their contents. The modifications are meant to improve the readability for readers without a background in TAS.
- Paragraph mentioning GPR (now fifth paragraph): We added a sentence on GPR for readers not familiar with Gaussian processes.
- Paragraph contrasting gpCAM (now seventh paragraph): We make clear what we mean when we claim that “gpCAM risks losing some of its interpretability”.

Comparison with gpCAM

- Last paragraph of the introduction: We added a sentence mentioning a comment in the Results section on a comparison with gpCAM.
- Benchmark subsection in Results section: We added a paragraph explaining why a comparison with gpCAM is currently difficult. However, we refer to the Supplementary Information where we perform a comparison that is currently possible based on all the information available to us about gpCAM. In particular, we choose an acquisition function that was used before for a neutron experiment at the instrument ThALES (ILL).
- Supplementary Information: We added a section for a comparison of the mentioned type with gpCAM.

General changes (not related to reviewers’ comments)

- We replaced all occurrences of “GPcam” by “gpCAM”, which is the correct variant (see `gpcam.1b1.gov`), and use a sans serif font for it so that it becomes gpCAM.
- We added a small paragraph at the end of our conclusions. It contains the description of another interesting topic for future research: a Bayesian estimation of model parameters using the autonomously collected data, if a model is available.
- We added a section “Data availability” and a corresponding data availability statement.
- We updated the commit SHAs of the repositories referenced in the section “Code availability”. The new commits however do not touch the functionality of our approach.
- We changed the number of instrument axes considered and their corresponding angles from 4 to 3 (Supplementary Information, Sec. 1). This fits better with the constant- k_f mode we used for our experimental setups.
- We fixed some entries in the table with milestone values for each benchmark test case (Supplementary Table 1). The former values were accidentally copied incorrectly.
- We made minor changes to wording, spelling, and grammar across the whole document.

References

- [1] M. M. Noack et al. “Gaussian processes for autonomous data acquisition at large-scale synchrotron and neutron facilities”. In: *Nature Reviews Physics* 3.10 (2021), pp. 685–697. DOI: 10.1038/s42254-021-00345-y.
- [2] M. M. Noack et al. “A kriging-based approach to autonomous experimentation with applications to X-ray scattering”. In: *Scientific Reports* 9 (2019), p. 11809. DOI: 10.1038/s41598-019-48114-3.
- [3] M. Teixeira Parente et al. “Benchmarking autonomous scattering experiments illustrated on TAS”. In: *Frontiers in Materials* 8 (2022), p. 772014. DOI: 10.3389/fmats.2021.772014.

REVIEWERS' COMMENTS

Reviewer #1 (Remarks to the Author):

The updated version of this manuscript reads very well. The authors addressed all my comments. I recommend publication.

Reviewer #2 (Remarks to the Author):

My concerns and questions have been addressed. I recommend acceptance as is.